# Endogenous modulation of human visual cortex activity improves perception at twilight

Lorenzo Cordani[1,2], Enzo Tagliazucchi[1,3,4], Céline Vetter[5,6], Christian Hassemer[1,7], Till Roenneberg[6], Jörg H. Stehle[7] & Christian A. Kell [1,2]

Perception, particularly in the visual domain, is drastically influenced by rhythmic changes in ambient lighting conditions. Anticipation of daylight changes by the circadian system is critical for survival. However, the neural bases of time-of-day-dependent modulation in human perception are not yet understood. We used fMRI to study brain dynamics during resting-state and close-to-threshold visual perception repeatedly at six times of the day. Here we report that resting-state signal variance drops endogenously at times coinciding with dawn and dusk, notably in sensory cortices only. In parallel, perception-related signal variance in visual cortices decreases and correlates negatively with detection performance, identifying an anticipatory mechanism that compensates for the deteriorated visual signal quality at dawn and dusk. Generally, our findings imply that decreases in spontaneous neural activity improve close-to-threshold perception.

[1] Cognitive Neuroscience Group, Brain Imaging Center, Goethe University, 60528 Frankfurt am Main, Germany. [2] Department of Neurology, Goethe University, 60528 Frankfurt am Main, Germany. [3] Brain and Spine Institute, Hôpital Pitié Salpêtrière, 75013 Paris, France. [4] Departamento de Física, Instituto de Física de Buenos Aires-CONICET, Buenos Aires 1428, Argentina. [5] Department of Integrative Physiology, University of Colorado, Boulder, CO 80310, USA. [6] Institute of Medical Psychology, Ludwig Maximilian University, 80336 Munich, Germany. [7] Institute of Anatomy III, Goethe University, 60590 Frankfurt am Main, Germany. Correspondence and requests for materials should be addressed to C.A.K. (email: c.kell@em.uni-frankfurt.de)

The day-night cycle leads to rhythmic changes in environmental conditions, particularly of ambient light. The circadian timing system anticipates such time of day (ToD)-dependent environmental rhythms and modulates physiology accordingly[1–3]. Animal models revealed ToD-dependent changes in genome readout, protein dynamics, and in electrophysiology that are regulated by the circadian system[3–5] and homeostatic factors[6]. However, the neural bases of ToD-dependent changes in perception and cognition in humans are unclear[7,8].

Perception can easily be studied in humans. An appropriate method to investigate the neural signatures of cognitive processes is the measurement of blood oxygen level dependent (BOLD) signal by functional magnetic resonance imaging (fMRI). fMRI during tasks can reveal brain activity associated with cognition, while fMRI during wakeful rest identifies more directly endogenous processes[9]. ToD modulates task-related BOLD-fMRI activity[10,11] but more importantly also functional connectivity in resting-state networks[12,13], suggesting that ToD modulates neural networks both during tasks but also independently of a cognitive task or sensory stimulation. Yet, the relationship between such network dynamics and ToD-dependent modulations of perception and behavior has not yet been shown. This lack of knowledge is surprising, because ToD dramatically influences the quality of the sensory input signals to the brain. Understanding the way ToD affects cortical dynamics may grant insights in how the brain deals with predictable challenges.

We investigated the ToD-dependency of BOLD signal variance – a quantification of the BOLD signal amplitude over time – as it provides important information about the relationship between brain activity and behavior[14,15]. BOLD signal variance was first analyzed during resting-state to detect endogenous ToD-dependent modulation of brain activity. In a second step, we investigated the consequences of such an endogenous modulation of cortical activity on visual perception. Our results reveal that BOLD signal variance drops endogenously at twilight compared to midday in sensory cortices while close-to-threshold visual perception improves in parallel. This identifies reductions in spontaneous sensory cortex activity as an anticipatory mechanism to compensate for the low sensory signal quality at twilight.

## Results

**Time-of-day-effect in resting-state BOLD signal variance.** Fourteen healthy male participants were scanned using fMRI at six different ToD on two subsequent days (08:00, 11:00, 14:00, 17:00, 20:00, 23:00 h on each day). This was done to account for habituation effects to the experimental setting. Each scanning session included a resting-state measurement in constant dim light (< 0.1 lux). Voxel-wise BOLD variance was defined in each individual by the resting-state BOLD signal standard deviation (SD) at each ToD. A whole-brain, repeated measures ANOVA (cluster-extent based threshold: $k = 395$, $P < 0.05$, family-wise error (FWE) corrected; cluster-defining primary threshold of $F_{(5, 65)} > 4.70$, $P < 0.001$, uncorrected; $n = 14$) revealed main effects of ToD on BOLD variance only in sensory cortices, namely, the bilateral visual ($P = 0.001$), somatosensory (three clusters, all $P = 0.001$), and right auditory cortex ($P = 0.007$, Fig. 1a and Table 1).

Post-hoc two-tailed dependent $t$-tests with Bonferroni correction (significance threshold at $P < 0.05$, $n = 14$) based on BOLD SD averages within each cluster and calculated separately for each ToD showed that this effect was driven by significant BOLD SD reductions at both 08:00 and 20:00 h compared to midday measurements 11:00, 14:00, and 17:00 h (Fig. 1b and Table 2 for statistical details). Reductions ranged from 17.9 to 25.8% of BOLD SD values at 14:00 h. As functional connectivity analyses have identified the meaningful part of resting-state BOLD activity

in a low frequency band around 0.01–0.1 Hz[16,17], we replicated the BOLD SD results in the frequency domain by calculating the amplitude of low-frequency fluctuations (Supplementary Fig. 1, Supplementary Tables 1 and 2). BOLD SD decreases in sensory cortices at 08:00 h and 20:00 h temporally coincided with local times of civil twilight, as the 08:00 h scan was close to or matched sunrise (end of civil morning twilight, range = 05:16–08:24 h), while the 20:00 h scan was close to or matched sunset (start of civil evening twilight, range 16:24–21:39 h). This suggests a particular role of the visual compared to the other sensory cortices in ToD-dependent modulation of brain activity. Note that the light conditions in the scanner were identical at each ToD and participants adapted to the dim light before measurement.

**Potential masking factors.** A range of masking factors could have potentially influenced our observation. Yet, the main effect of ToD on BOLD SD in the visual cortex remained significant after accounting for potential confounders, including heart rate, breathing rate, body temperature, subjective sleepiness, and the amplitude and variance of head motion parameters, as well as the number of outliers in head motion parameters in a linear mixed model (main effect of ToD $F_{(5, 34.029)} = 4.177$, $P = 0.005$, type III $F$-test, $n = 14$). This model also included chronotype, sleep pressure, sleep debt, and scanning days to investigate effects of interindividual differences (see section Circadian and homeostatic factors) and habituation on resting-state BOLD SD. The repetitive measurements on two subsequent scanning days did not introduce habituation effects, because ToD remained significant when accounting for scanning days in the model and there was no significant interaction between ToD and scanning day ($F_{(5, 41.829)} = 0.871$, $P = 0.509$, type III $F$-test, $n = 14$).

**Vigilance.** Because the main effect of ToD in BOLD SD remained significant when adjusting for subjective sleepiness, it is likely that the ToD-dependent modulation is not explained by vigilance states. We additionally investigated whether an objective marker of EEG-documented vigilance changes[18,19] showed comparable ToD-effects. We tested the main effect of ToD in resting-state thalamo-cortical connectivity using correlation maps of thalamic BOLD time series with all other brain voxels. This parameter was not affected by ToD (whole-brain repeated measures ANOVA, cluster size $k = 3$, $P > 0.999$, FWE cluster correction, $n = 14$) suggesting that both subjective and objective measures of vigilance states do not explain the diurnal modulation in resting-state BOLD SD in the sensory cortices. To ascertain that this negative finding did not result from insufficient statistical power, we performed a bootstrap analysis using a published dataset in which vigilance changes have been investigated[18], testing whether a group size of 14, as used in the present study, is sufficient to detect vigilance changes. Indeed, vigilance-related changes in thalamo-cortical connectivity can reliably ($1–ß = 0.8$) be detected in sample sizes as the one used in this study.

**Circadian and homeostatic factors.** The lack of an association with vigilance state rather suggests an endogenous modulation of resting state cortical activity by circadian and/or homeostatic factors. When we examined the region of the suprachiasmatic nucleus in the hypothalamus[20], which is the central pacemaker of the circadian system in the brain, we did not observe a modulation of resting-state BOLD variance as a function of ToD (repeated measures ANOVA, all voxels $F_{(5, 65)} < 4.25$ and $P \geq 0.09$, small volume corrected, $n = 14$). Still, it remains possible that the ToD effects in BOLD SD in sensory cortices are regulated by the circadian system, because the suprachiasmatic nucleus codes time at each ToD in the magnitude of firing rate rather than

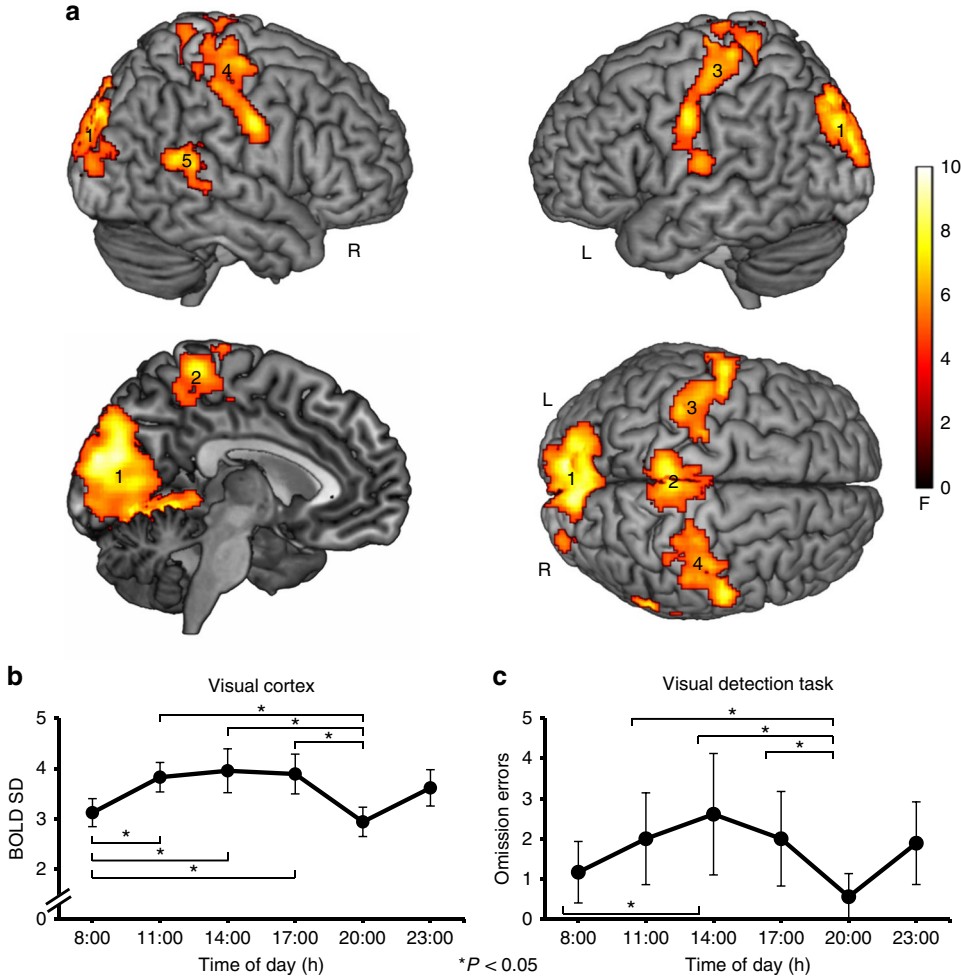

**Fig. 1** Resting-state BOLD SD is reduced in sensory cortices at twilight, when visual detection is improved. **a** Significant main effect of ToD for resting-state BOLD SD in visual (cluster 1, peak MNI value −8 −90 20), somatosensory (clusters 2–4, peak MNI values −4 −38 68; −60 −8 40; 58 −8 36), and right auditory cortices (cluster 5, peak MNI value 64–48 20) (cluster-extent based threshold: $k = 395$, $P < 0.05$, family-wise error (FWE) corrected; cluster-defining primary threshold of $F(5, 65) > 4.70$, $P < 0.001$, uncorrected; $n = 14$). Color indicates $F$-values. For further statistical details see Table 1. **b** Group analysis showing significant resting-state BOLD SD decreases in visual cortex (cluster 1) at 08:00 and 20:00 h compared to midday measurements (graph shows the group-mean BOLD SD values and error bars represent the 95% confidence interval. Asterisks mark significant pairwise differences at $P < 0.05$, Bonferroni corrected). Note that somatosensory, and right auditory cortex show similar ToD-effects (Supplementary Fig. 1c). For further statistical details see Table 2. **c** Main effect of ToD for omission errors during the visual detection task. Participants committed significantly less omission errors at the same times, when resting-state BOLD SD was decreased (see **b**; graph shows the group-mean omission errors and error bars represent the 95% confidence interval. Asterisks mark significant pairwise differences at $P < 0.05$)

in firing rate variance[21,22]. Indeed, both, chronotype, a proxy for circadian phase, and sleep pressure, a homeostatic factor, explained part of the ToD modulation in BOLD variance (Supplementary Fig. 2). The linear mixed model on resting-state BOLD SD in the visual cortex showed a significant interaction between ToD and chronotype ($F(5, 34.603) = 2.738$, $P = 0.037$, type III $F$-test, $n = 14$) and between ToD and sleep pressure ($F(5, 32.21) = 5.128$, $P = 0.001$, type III $F$-test, $n = 14$). There was no significant interaction between ToD and sleep debt ($F(5, 38.554) = 1.922$, $P = 0.113$, type III $F$-test, $n = 14$).

While our study was not originally designed to dissociate the specific contribution of circadian and homeostatic factors in the ToD-dependent modulation of sensory cortex activity, a sleep deprivation study in a larger sample documented interactions between circadian and homeostatic influences and BOLD activity during a visual psychomotor vigilance task[10]. In this prior study, an interaction between homeostatic and circadian factors was particularly observed in posterior, mostly occipital cortices, while the circadian modulation seemed less important in the frontal

cortices. This observation suggests that the circadian system modulates perceptual processes in the sensory cortices and could potentially be the primary mediator of the ToD effects observed here.

**Time-of-day-effects in close-to-threshold visual perception.** Because the ToD-dependent modulation of BOLD SD represents changes in the spontaneous neural activity during resting-state, independent of sensory stimulation and task demands, our results suggest a ToD-dependent endogenous regulation of cortical activity. To reveal the relationship between such neural ToD effects in sensory cortices and perception we examined ToD effects on close-to-threshold visual perception, because dawn and dusk affect the signal quality of primarily the visual input. During all measurements, participants performed also a close-to-threshold visual detection task while still being scanned in constant dim light. A repeated measures ANOVA showed a significant main effect of ToD on the number of omission errors ($F$

**Table 1 Brain regions showing ToD-effects of resting-state BOLD variance**

| Anatomical region | Cluster P-value (FWE) | Cluster size(in voxels) | Local maxima | MNI coordinates (x, y, z) |
|---|---|---|---|---|
| Cluster 1 Occipital cortex (visual cortex) | 0.001 | 9089 | L Cuneus | −8, −90, 20 |
| | | | L Cuneus | −8, −80, 22 |
| | | | R Lingual gyrus | 10, −70, −6 |
| | | | R Calcarine gyrus | 12, −68, 4 |
| | | | R Cuneus | 8, −88, 32 |
| | | | R Cuneus | 6, −84, 28 |
| | | | R Cuneus | 8, −86, 24 |
| | | | R Lingual gyrus | 20, −60, −4 |
| | | | L Cuneus | −4, −80, 30 |
| | | | L Lingual gyrus | −10, −74, −6 |
| | | | L Lingual gyrus | −16, −52, −4 |
| | | | L Superior occipital gyrus | −16, −82, 26 |
| | | | R Cuneus | 2, −86, 16 |
| | | | R Calcarine gyrus | 10, −78, 16 |
| | | | L Calcarine gyrus | −10, −66, 8 |
| Cluster 2 Paracentral lobule (mesial somatosensory cortex) | 0.001 | 1332 | L Paracentral lobule | −4, −38, 68 |
| | | | R Paracentral lobule | 6, −26, 60 |
| | | | L Paracentral lobule | −4 −26, 80 |
| | | | R Paracentral lobule | 8, −44, 74 |
| | | | L Precuneus | −8, −48, 58 |
| Cluster 3 Left Rolandic cortex (lateral somatosensory cortex) | 0.001 | 1025 | L Postcentral gyrus | −60, −8, 40 |
| | | | L Postcentral gyrus | −56, −8, 42 |
| | | | L Postcentral gyrus | −50, −10, 40 |
| | | | L Postcentral gyrus | −36, −32, 70 |
| | | | L Postcentral gyrus | −46, −12, 38 |
| | | | L Precentral gyrus | −26, −20, 76 |
| | | | L Postcentral gyrus | −44, −22, 64 |
| | | | L Postcentral gyrus | −48, −20, 60 |
| | | | L Postcentral gyrus | −48, −18, 56 |
| | | | L Postcentral gyrus | −64, −16, 16 |
| | | | L Precentral gyrus | −46, −6, 26 |
| | | | L Precentral gyrus | −24, −24, 64 |
| | | | L Postcentral gyrus | −44, −12, 48 |
| Cluster 4 Right Rolandic cortex (lateral somatosensory cortex) | 0.001 | 1007 | R Postcentral gyrus | 58, −8, 36 |
| | | | R Precentral gyrus | 44, −16, 66 |
| | | | R Precentral gyrus | 30, −24, 72 |
| | | | R Postcentral gyrus | 50, −28, 56 |
| | | | R Postcentral gyrus | 50, −22, 50 |
| | | | R Postcentral gyrus | 30, −34, 72 |
| | | | R Postcentral gyrus | 26, −28, 64 |
| | | | R Precentral gyrus | 22, −28, 62 |
| | | | R Precentral gyrus | 34, −26, 60 |
| | | | R Postcentral gyrus | 40, −38, 62 |
| | | | R Precentral gyrus | 40, −18, 54 |
| Cluster 5 Right lateral temporal cortex (auditory cortex) | 0.007 | 395 | R Superior temporal gyrus | 64, −48, 20 |
| | | | R Superior temporal gyrus | 54, −40, 14 |
| | | | R middle temporal gyrus | 58, −50, 16 |
| | | | R middle temporal gyrus | 60, −40, 6 |
| | | | R Rolandic operculum | 44, −34, 22 |
| | | | R Superior temporal gyrus | 68, −34, 6 |
| | | | R Supramarginal gyrus | 46, −38, 24 |
| Left lateral temporal cortex (auditory cortex) | 0.084[a] | 206 | L Superior temporal gyrus | −66, −42, 16 |

Denomination of local maxima was adopted from the SPM Anatomy toolbox[47]. MNI-coordinates represent peak activations inside clusters
[a]Note that the left auditory cortex was slightly subthreshold

$(5, 40) = 2.67$, $P = 0.036$; partial eta-squared $= 0.250$, $n = 9$; Fig. 1c). Participants performed best at 08:00 h (mean ± SD omission errors $= 1.17 \pm 1.17$) and 20:00 h (mean ± SD omission errors $= 0.56 \pm 0.88$) compared to midday (omission errors all ≥ 2.00; Table 3). Reaction times were not significantly different throughout the day ($F(5, 40) = 2.01$, $P = 0.097$; partial eta-squared $= 0.201$, $n = 9$), indicating that omission errors are a better measure for detection of brief stimuli than reaction times.

Good visual detection performance at 08:00 and 20:00 h (Fig. 1c) occurred in parallel to low resting-state BOLD SD

(Fig. 1b), both coinciding with times of twilight (Pearson correlations accounting for the dependence among the repeated measures: $r = 0.54$, $P < 0.001$, $n = 54$). We observed a positive, linear relationship between resting-state BOLD SD in the visual cortex and omission errors during visual detection at the same times of day (i.e., low resting-state BOLD SD was associated with good visual detection, Fig. 2a). This relationship between behavior during the visual detection task and prior task-independent resting-state activity points towards an anticipatory mechanism that facilitates visual perception. Yet, linking BOLD

**Table 2 Repeated measures ANOVA post-hoc *t*-tests on resting-state BOLD SD**

| Anatomical region | Post-hoc *t*-test | *t*(13) | *P*-value (Bonf.) | Cohen's *d* |
|---|---|---|---|---|
| Cluster 1 Visual cortex | 11:00–08:00 h | 4.68 | 0.006 | 1.29 |
| | 14:00–08:00 h | 4.05 | 0.021 | 1.15 |
| | 17:00–08:00 h | 3.95 | 0.025 | 1.15 |
| | 11:00–20:00 h | 4.84 | 0.005 | 1.59 |
| | 14:00–20:00 h | 4.01 | 0.022 | 1.43 |
| | 17:00–20:00 h | 5.04 | 0.003 | 1.41 |
| Cluster 2-4 Somatosensory cortex | 11:00–08:00 h | 3.97 | 0.024 | 0.90 |
| | 14:00–08:00 h | 3.75 | 0.037 | 0.87 |
| | 17:00–08:00 h | 3.90 | 0.028 | 0.94 |
| | 11:00–20:00 h | 3.68 | 0.042 | 1.21 |
| | 14:00–20:00 h | 3.81 | 0.033 | 1.12 |
| | 17:00–20:00 h | 4.21 | 0.015 | 1.17 |
| Cluster 5 Right auditory cortex | 11:00–08:00 h | 4.50 | 0.009 | 0.64 |
| | 14:00–08:00 h | 5.03 | 0.003 | 0.69 |
| | 17:00–08:00 h[a] | 3.06 | 0.138 | 0.56 |
| | 11:00–20:00 h | 5.31 | 0.002 | 1.10 |
| | 14:00–20:00 h | 4.39 | 0.011 | 1.01 |
| | 17:00–20:00 h | 4.58 | 0.008 | 0.90 |

Paired samples *t*-tests for BOLD SD for each cluster (significance threshold at *P* < 0.05, Bonferroni corrected). The *t*-Test marked with "a" was subthreshold, but listed in the table for completeness. All other comparisons were *P* < 0.05

**Table 3 Repeated measures ANOVA post-hoc *t*-tests on omission errors**

| Post-hoc *t*-test | Mean Δ in omission errors | *t*(8) | *P*-value | Cohen's *d* |
|---|---|---|---|---|
| 11:00–08:00 h[a] | 0.83 | 2.00 | 0.081 | 0.52 |
| 14:00–08:00 h | 1.44 | 2.76 | 0.025 | 0.60 |
| 17:00–08:00 h[a] | 0.83 | 1.40 | 0.199 | 0.54 |
| 11:00-20:00 h | 1.44 | 2.69 | 0.027 | 0.98 |
| 14:00-20:00 h | 2.06 | 3.26 | 0.012 | 0.94 |
| 17:00-20:00 h | 1.44 | 2.83 | 0.022 | 0.91 |

Planned paired samples *t*-tests based on the results of the analysis of BOLD SD (significance threshold at *P* < 0.05). *t*-Tests marked with "a" were subthreshold, but listed in the table for completeness. All other comparisons were *P* < 0.05

variance reductions to performance requires correlating not only resting-state, but also task-related BOLD variance with visual detection performance.

**ToD-effects in BOLD signal variance during visual perception.** Therefore, we further analyzed fMRI data during the visual detection task. We first tested whether ToD modulates task-related BOLD SD in the visual cortex during visual detection in the same way as it modulates visual cortex BOLD SD during rest using a *t*-test (cluster-extent based threshold: $k = 112$, $P < 0.05$, FWE corrected; cluster-defining primary threshold of $t(40) > 3.31$, $P < 0.001$, uncorrected; $n = 9$). Task-related BOLD SD in two clusters in the primary and secondary visual cortex was reduced at 8:00 and 20:00 h, as compared to midday measurements (Fig. 3a and b). Reductions ranged from 7.4 to 26.8% of BOLD SD values at 14:00 h. In comparison to resting state, BOLD SD during the visual detection task was overall decreased in both the primary and secondary visual cortex (Rest-task: $t(8) = 3.82$, $P = 0.010$, Cohen's $d = 0.99$, mean difference = 1.26, $n = 9$) and the secondary visual cortex (Rest-task: $t(8) = 4.75$, $P = 0.002$, Cohen's $d = 1.30$, mean difference = 1.63, $n = 9$); significance threshold was at $P < 0.05$, Bonferroni corrected. Task-related BOLD SD in

both visual cortex clusters was also linearly and positively related to omission errors during visual detection at the same ToD (Pearson correlations accounting for the dependence among the repeated measures in primary and secondary visual cortex: $r = 0.43$, $P = 0.003$, $n = 54$ and secondary visual cortex: $r = 0.42$, $P = 0.004$, $n = 54$; Fig. 2b). The similar ToD-dependent effects during, both, the visual detection task and rest, suggest that endogenous reductions in spontaneous cortical activity persist during active processing.

## Discussion

Here we report reduced visual cortex BOLD signal variance during close-to-threshold perception at times of twilight that correlates with improved visual detection. Because activity is also reduced during prior resting-state and masking factors have been accounted for in our analyses, an endogenous source of this ToD-dependent modulation of cortical activity is likely.

Neuromodulatory systems such as the acetylcholine-basal forebrain system or the norepinephrine-locus coeruleus system are known to be under circadian influence[23,24] and to modulate homeostatic processes[6]. Both noradrenergic and cholinergic systems impact neural processing in sensory cortices by increasing the signal to noise ratio (SNR) and consequently facilitate processing of weak incoming signals[25,26]. Neuromodulators could thus potentially mediate both, the SNR enhancements and ToD-dependency of endogenous resting-state and perception-related BOLD activity.

Although previous work has shown stimulus-induced decreases in BOLD variance when comparing task conditions to rest[15,27], our findings are novel in that they demonstrate BOLD variance reductions, occurring even in the absence of any external stimuli or task conditions. While endogenous fluctuations in neural activity in sensory cortices have been associated with subsequent visual and auditory perception[28], this study reveals a ToD-dependency of such relationship between endogenous BOLD variance and perception. A direct facilitatory role of BOLD variance reductions in visual perception is likely given the consistent association between visual perception and variance in visual cortex activity during both rest, and visual detection. We propose that this anticipatory reduction of ongoing neural activity in visual cortices at times of twilight represents a mechanism to increase the SNR for close-to-threshold visual perception.

In addition to ToD-dependent reductions of spontaneous neural activity in human visual cortices, we observed the same effect in somatosensory and auditory cortices. Such a parallel increase in SNR in these cortices may further enhance perception in general, since multisensory interactions improve the accuracy of sensory processing when vision alone provides insufficient sensory evidence[29,30]. Indeed, an increased auditory stimulus discrimination performance in humans has been described to occur at early and late ToD (around 06:00 and 20:00 h)[31].

From a dynamical systems perspective, BOLD variance can be interpreted as a "wandering" of the brain around attractor states[32,33]. Within this framework, our results suggest that spontaneous exploration of internal brain states, involving sensory areas, is reduced at times of twilight, which in turn could facilitate detection of weak external stimuli. Because we did not investigate time points after 23:00 h, we cannot determine whether the observed BOLD variance reductions occur isolated at times of twilight or whether they also occur during later phases of the night. In addition, we did not measure light exposure prior to scanning and calorie intake, so that we could not examine the relationship between these factors and BOLD SD.

Humans, a day-active species, largely rely on the visual sense for spatial orientation. Yet, even in ancestral societies, human activity usually extended into morning and evening times of twilight, when natural illuminance is drastically reduced[34]. An

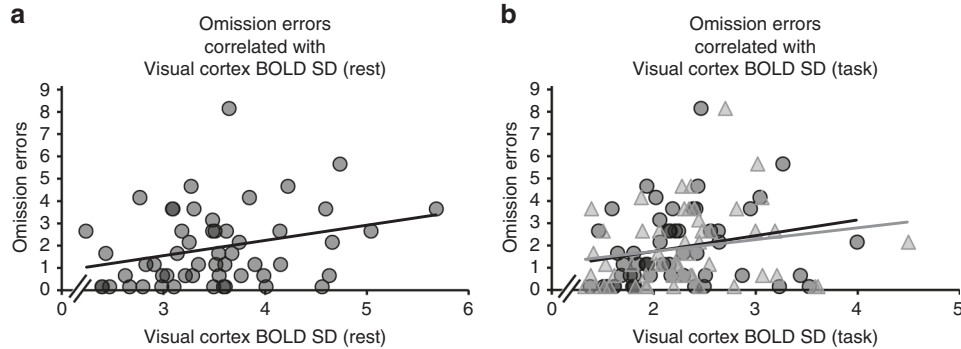

**Fig. 2** BOLD variance is positively associated with omission errors during visual detection. Pearson correlations that accounted for the dependence among the repeated measures, were calculated to study the relationship between behavior and BOLD SD in visual cortex during rest (**a**) and task (**b**). **a** Individual resting-state BOLD SD values in visual cortex (extracted from cluster 1, $x$-axis) correlated positively with individual omission errors during visual detection ($y$-axis) at the same times of day ($r = 0.54$, $n = 54$, $P < 0.001$). The scatter plot shows individual resting-state BOLD SD values and corresponding omission errors for all ToD (linear regression line of best fit included in the graph). **b** Correlation between omission errors and BOLD SD persisted during visual detection, linking BOLD SD reductions to performance ($y$-axis: omission errors; $x$-axis: BOLD SD during visual detection task). BOLD SD values were extracted from the cluster in the primary and secondary visual cortex (circles: $r = 0.43$, $P = 0.003$, $n = 54$. Local maxima in the left calcarine sulcus (MNI −10 −68 12) and left lingual gyrus (MNI −8 −66 2), see Fig. 3) and from the cluster in the secondary visual cortex (triangles: $r = 0.42$, $P = 0.004$, $n = 54$. Local maxima in the left (MNI −2 −82 24) and right cuneus (MNI 4–82 20), see Fig. 3). The scatter plot shows individual BOLD SD values during visual detection and corresponding omission errors for all ToD (linear regression line of best fit included in the graph)

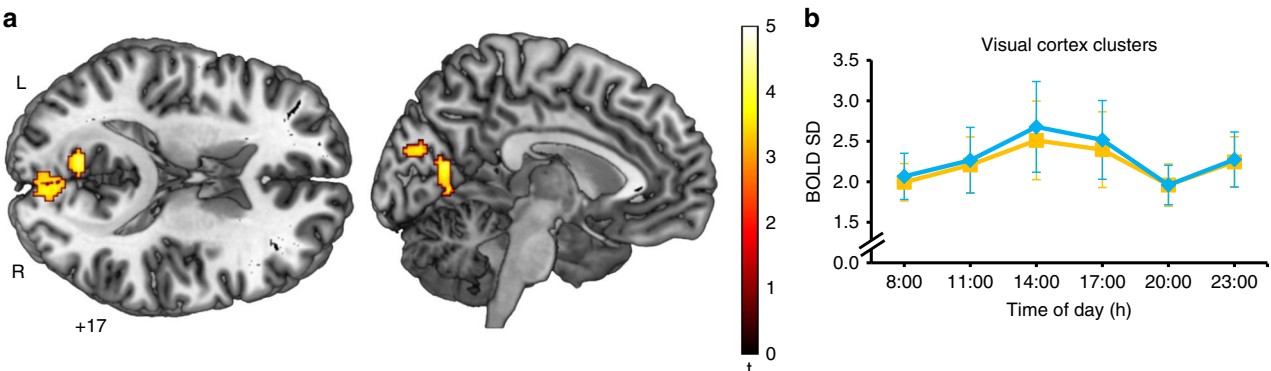

**Fig. 3** BOLD SD during visual detection decreases in visual cortex at twilight. **a** One-sample $t$-test for BOLD SD during visual detection showing significant BOLD SD reductions in a cluster in the primary and secondary visual cortex ($k = 112$ voxels, $P = 0.022$, FWE; local maxima in the left calcarine sulcus (MNI −10 −68 12) and left lingual gyrus (MNI −8 −66 2)) and a cluster in the secondary visual cortex ($k = 184$ voxels, $P = 0.006$, FWE; local maxima in the left (MNI −2–82 24) and right cuneus (MNI 4 −82 20)) at 08:00 h and 20:00 h compared to midday measurements 11:00, 14:00, and 17:00 h (cluster-extent based threshold: $k = 112$, $P < 0.05$, FWE corrected; cluster-defining primary threshold of $t(40) > 3.31$, $P < 0.001$, uncorrected; $n = 9$). **b** Extracted task-related BOLD SD values from the cluster in the primary and secondary visual cortex (yellow) and from the cluster in secondary visual cortex (blue), showing the same ToD-effects as resting-state BOLD SD. Pairwise comparisons between all ToD were not performed on the extracted BOLD SD values to avoid an inflation in Type I error[56]. The graph shows the group-mean BOLD SD values for both clusters

anticipatory increase in the SNR in the visual system during these times, compensating for the environmental visual constraints, may thus have been crucial for survival until the introduction of electric light sources into social life[35]. Beyond this vital importance, our findings may represent a new generic mechanism underlying improved perception of close-to-threshold sensory input.

## Methods

**Participants**. Fourteen healthy male volunteers were included in this study (mean age = 23.8, range = 20–27). Four additional participants were recruited for this study, but had to be excluded from analysis because of non-completion of the entire study protocol or non-correctable imaging artifacts. To enable a quantification of potential habituation effects, participants came in on two days, and were assessed multiple times throughout the day. Because of this resource- and cost-intense study design, sample size was limited. No participant had a history of neuropsychiatric or vascular disease, took medication or psychoactive substances, had recent or actual sleep disturbances, worked shifts, or consumed caffeine excessively (>3 cups of coffee or other caffeinated beverages per day). We restricted this study to male participants because of reported gender variations in resting-

state dynamics[36] and possible interactions between the menstrual cycle and circadian rhythms[37]. All participants gave their informed consent and received payment for their participation. This study has been approved by the ethics commission of the Medical Faculty of the Goethe University Frankfurt (GZ 244/09) in accordance with the Declaration of Helsinki.

We instructed all participants to follow regular bedtimes and wake-up-times (between 23:00–00:00 h and 7:00–8:00 h, respectively) for five days before scanning until completion of the experimental phase. Compliance was monitored with a diary of activities and by continuous wrist-actigraphy recordings[38]. The actigraph (Actiwatch Mini; CamNtech Ltd., Papworth Everard, UK) was worn at all times on the non-dominant wrist except during scanning sessions. Individual differences in the phase of entrainment – i.e., the chronotype[39] – and increasing homeostatic sleep pressure during time spent awake, but also increasing sleep debt as a marker of sleep deprivation, are known to modulate the ToD-dependent profile of neurobehavioral variables[7].Therefore, the participants' individual chronotype was assessed with the Munich Chronotype Questionnaire and characterized with a single value, the midpoint of sleep on free days, corrected for sleep debt accumulated during work days (MSFsc)[39–41]. We a priori excluded extreme chronotypes. The mean ± SD MSFsc of the participants was 4.87 ± 0.92 (range = 3.89–6.71). The MSFsc could not be assessed in two participants due to non-completion of the questionnaire.

Individual wake-up time on scan days was extracted from actigraphy and used as an indicator for sleep pressure in each participant at each ToD[38,42]. Since participants were always scanned at the same times, an earlier wake-up time indicates higher sleep pressure at each ToD compared to later wake-up times. Mean ± SD wake-up time of the participants was 06:10 h ± 00:39 h (range = 05:00–07:10 h). Wake-up time could not be calculated in one participant, because actigraphy could not be obtained on scan days. We estimated the participants' sleep debt, indexed by the difference between mean weekly sleep duration from the Munich Chronotype Questionnaire and sleep duration during scanning days extracted from actigraphy[40–42]. Because of non-completion of the questionnaire and missing actigraphy data (see above), sleep debt could not be assessed in three participants. Mean ± SD sleep debt of the participants was 1.81 ± 1.00 (range = 0.17–3.09). Higher values indicate higher sleep debt. Because actigraphy-based sleep estimates often under-estimate the duration of sleep, as compared to self-reports[43], this quantification possibly overestimates the true extent of sleep pressure in the population.

Before each fMRI session, body temperature was measured with an ear thermometer and the participants' sleep propensity was assessed with a self-report questionnaire based on a modified version of the Epworth Sleepiness Scale (ESS) questionnaire[44], in which participants were asked to rate their sleep propensity by projecting their actual vigilance state in the situations provided by the questionnaire to quantify diurnal sleepiness. Participants did not nap or consume stimulants during scanning days. Time in this paper is expressed in Central European Time (CET), the German local time. Times of twilight were obtained from the German Meteorological Office.

**Resting-state fMRI.** The participants underwent six fMRI scanning sessions at fixed times on two subsequent days (8:00, 11:00, 14:00, 17:00, 20:00, and 23:00 h on each day) to account for habituation effects to the experimental setting. To minimize interferences with the participants' sleep, no measurements were scheduled between 24:00 and 07:00 h. Each fMRI session included a 7 min resting-state measurement in dim light. The participants were instructed to lie still and comfortably in a supine position, to keep their eyes open and fixate a white crosshair on a black background (< 1 lux, maximum illuminance measured at corneal level using a VOLTCRAFT LX-1108 lux meter; Voltcraft, Hirschau, Germany), not to fall asleep, and not to think of anything in particular and let their mind wander freely. A minimum of five minutes elapsed between switching off the main lights in the scanning room and the beginning of the functional measurement, allowing participants to adapt to the dim light conditions. Participants wore headphones and earplugs to protect against the scanner noise, and their heads were immobilized with foam cushions to minimize movement artifacts. If visual correction was necessary, participants were provided with appropriate MRI compatible corrective lenses. A coil-mounted mirror allowed viewing the projection screen and the white crosshair on a black background was displayed using Presentation software (Neurobehavioral Systems Inc., Berkely, CA, USA; http://www.neurobs.com).

During fMRI scans, we monitored the participants visually and recorded their cardiorespiratory and movement parameters. Prior to the experimental phase, participants underwent an 'acclimatization session' in the scanner to acclimatize to the unfamiliar environment and experimental procedures.

**Visual detection task fMRI.** Nine of the original participants additionally performed a visual detection task in dim light while still being scanned for additional 4 min 52 s at each ToD. They were instructed to press a button with their right index finger as soon and as quickly as possible when they saw a low-contrast orange crosshair (< 1 lux) flashing shortly (500 ms) on the center of a black screen inside the scanner (background light in the dimly lit scanner room was <0.1 lux). Three one minute task blocks with 11 crosshair presentations each (randomized interstimulus interval from 2–10 s) alternated with four 25 s fixation blocks (white crosshair, <1 lux, as during resting-state measurements).

**Data acquisition and preprocessing of resting-state fMRI data.** Resting-state data were collected using a 3 Tesla MRI-scanner (Siemens MAGNETOM Allegra, syngo MR 2004A, Erlangen, Germany). A total of 12 resting-state sessions of a gradient-echo T2*-weighted transverse echo-planar imaging sequence (EPI) were acquired for each participant. A resting-state session comprised 210 EPI volumes (6 sessions on two subsequent days = 12 × 210 volumes for each participant). Each volume contained 30 axial slices acquired in a sequential manner, covering the whole brain (TR/TE/flip angle = 2000ms/30 ms/90°, FOV = 192 mm, matrix size (resolution) = 64 × 64, voxel size of 3 × 3 × 3 mm³, distance factor 25%). The first 4 volumes were discarded to avoid magnetic saturation effects. Spatial preprocessing was performed using standard algorithms implemented in SPM 12 (Statistical Parametric Mapping 12, Version 6225; Wellcome Trust Center for Neuroimaging, London, UK; http://www.fil.ion.ucl.ac.uk/spm/): head-motion correction via intra-subject spatial realignment (least squares approach and 6 parameter rigid body spatial transformation), spatial normalization to the standard EPI template of the Montreal Neurological Institute (MNI) and resampling to a voxel size of 2 × 2 × 2 mm³ (using a 12 parameter affine transformation), and spatial smoothing with an isotropic 8 mm full-width at half maximum (FWHM) Gaussian kernel[45].

Resting-state BOLD data are contaminated by non-neural signals originating from residual motion artifacts, respiration, cardiac beats, and scanner drifts, which makes additional preprocessing steps necessary[46]. Using in-house scripts in

MATLAB 7.7 (The MathWorks, Inc., Natick, MA, USA; https://www.mathworks.com/), data were first high pass filtered with a 0.01 Hz cutoff using a 6th order Butterworth filter. Then, in order to remove the global effect of the various noise sources, the six motion parameters computed in the realignment step and fMRI time series from cerebrospinal fluid (CSF) and white matter (WM), as well as the first derivatives of each, were detrended and regressed out of the resting-state data using least squares linear regression. The CSF and WM time series (CSF: averaged within a 2 mm sphere at MNI coordinates −8, −8, 21, WM: averaged within a 5 mm sphere at MNI coordinates 30, −20, 30) were extracted using the MarsBaR toolbox (Version 0.44; http://marsbar.sourceforge.net). Finally, a mask based on the Automated Anatomical Labeling (AAL) atlas[47] was applied to all volumes to restrict further analysis to gray matter-voxels only.

A high-resolution T1-weighted 3D magnetization-prepared rapid acquisition gradient echo (3D MP-RAGE) sequence was also acquired for each participant (144 sagittal slices, 1 slab, TR/TE/TI/flip angle = 2250 ms/2.6 ms/900 ms/9°, FOV = 256 mm, matrix size (resolution) = 224 × 256, voxel size 1 × 1 × 1.10 mm³, distance factor 50%). The high-resolution anatomical T1 sequence was used to exclude any structural anomalies and to map the functional results onto an averaged brain with higher anatomical resolution (see Supplementary Fig. 1).

**Single subject analyses of resting-state fMRI data.** Three variables of interest were calculated from the resting-state BOLD time series for each session (ToD) separately, using in-house scripts in MATLAB 7.7. Whole brain, voxel-wise standard deviation (SD) of the BOLD signal: We calculated the voxel-wise SD of the BOLD time series for each ToD, resulting in an SD-map for each ToD. We obtained voxel-wise SD with:

$$SD = \sqrt{\frac{\sum_{i=1}^{n}(x_i - \bar{x})^2}{n-1}}$$

$\{x_1, x_2, …, x_n\}$ represents the single values of a time series of length $n$ and $\bar{x}$ is the mean value of the whole time series.

Whole brain, voxel-wise amplitude of low-frequency fluctuation (ALFF) of the BOLD signal: The ALFF of the BOLD signal represents the power spectral density within the frequency band of 0.01–0.1 Hz[48,49]. Since the power spectrum of a time series describes how the variance of the data is distributed over the frequency domain, BOLD ALFF is analogous to BOLD SD in the given frequency range. We first transformed the BOLD time series into the frequency domain with the fast Fourier transform algorithm and then obtained BOLD ALFF by averaging the resulting power spectrum across the frequency band of interest[50]. Functional connectivity of the thalami: BOLD time series from the left and right thalamus, as defined by the AAL atlas, were extracted and averaged within each cluster for each ToD. The time series was correlated (Pearson's product-moment correlation coefficient) with each voxel separately, resulting in a functional connectivity map of correlation coefficients for each ToD[18].

**Group analysis of resting-state fMRI data.** Group statistical analyses were performed using SPM 12. First, for each participant, the BOLD SD, ALFF, and thalamic connectivity values acquired on the two consecutive days were averaged voxel-wise at each ToD separately, using the ImCalc tool in SPM, yielding BOLD SD, ALFF and thalamic connectivity whole brain maps for each of the six time points of interest. Then we used whole-brain, random-effects, one-way repeated measures ANOVAs (ToD as independent variable with 6 levels, homoscedasticity and sphericity were assumed) for group inference and tested for a main effect of ToD with an F-test, for BOLD SD, ALFF and thalamic connectivity values, respectively. All statistical brain maps were corrected for multiple comparisons by using cluster-extent based thresholding at $P < 0.05$, FWE. The cluster-defining threshold was set at $P < 0.001$ (uncorrected). We also report the MNI coordinates of local maxima in these clusters. The denomination of local maxima was adopted from the SPM Anatomy toolbox[51]. To identify the ToD-differences driving the main effect of ToD in BOLD SD, we performed additional post-hoc t-tests. First, BOLD SD values from significant clusters were averaged within each cluster and extracted for each ToD and participant separately using MarsBaR. Then, using IBM SPSS Statistics (Version 22; IBM, Armonk, NY, USA; https://www.ibm.com/analytics/us/en/technology/spss/), we performed a repeated measures ANOVA with subsequent pairwise comparisons between all ToD (dependent, paired samples, two-tailed t-tests with Bonferroni correction, significance threshold at $P < 0.05$) for each cluster. Clusters in right, left and mesial somatosensory regions were considered as one. BOLD ALFF values from significant clusters were extracted in the same way as for BOLD SD and used for planned pairwise comparisons based on the results of the analysis of BOLD SD (difference between 08:00 and 20:00 h and, midday measurements 11:00, 14:00, and 17:00 h). Again, clusters in right, left and mesial somatosensory cortex were considered as one. Clusters in left and right visual cortex were also considered as one. The three clusters in the left temporal cortex were also considered as one. Thalamic connectivity values were not investigated further, because we found no significant main effect of ToD in the whole-brain, repeated measures ANOVA. Cohen's $d$ statistic for repeated measures was calculated after Dunlap et al.[52]. Functional results of the BOLD SD analyses were rendered on the ch2better.nii template using MRIcron (Version 4 August 2014;

http://www.mccauslandcenter.sc.edu/mricro/mricron/). For visualization purposes, the high-resolution T1 images (only 13 participants, one participant failed to appear for the recording of the MP-RAGE sequence) were segmented and normalized in SPM 12, using the standard MNI templates. Then, using ImCalc, we averaged the T1-images and skull stripped the averaged image by masking with the gray-matter and white-matter images from the segmentation step. Functional results for Supplementary Fig. 1 were rendered on the averaged T1-image using MRIcroGL (Version 1 Jan 2015; http://www.mccauslandcenter.sc.edu/mricrogl/).

**Multi-variable adjustment of linear mixed models**. Resting-state activity can be affected by a range of potential confounders. To test whether the observed ToD effects in BOLD SD were driven by such effects, we calculated a linear mixed model with resting-state BOLD SD in the visual cortex as dependent variable. This model included ToD and scanning day as repeated measures fixed effects (autoregressive covariance structure AR(1)), as well as heart rate, breathing rate, body temperature, subjective sleepiness, amplitude and variance of head motion parameters, and the number of outliers in head motion parameters (defined as values above the mean ± 2,5 standard deviations), chronotype, sleep pressure, and sleep debt as fixed effects. Subjects were included as a random effects variable. We used the restricted maximum likelihood estimation method and tested whether the main effect of ToD in BOLD SD in the visual cortex remained significant ($P < 0.05$), when including all these potential confounding factors. To study whether chronotype, sleep pressure, and sleep debt, as well as scanning day explained part of the ToD modulation in BOLD variance, we additionally included multiplicative interaction terms in the multi-variable adjusted, linear mixed model. Interactions were considered significant at $P < 0.05$. A significant interaction between ToD and scanning day could potentially reveal habituation effects due to repetitive measurements on two subsequent scanning days. Yet, the interaction was not significant ($F(5, 41.829) = 0.871$, $P = 0.509$, type III $F$-test, $n = 14$).

**Region of interest analysis in the suprachiasmatic region**. We additionally investigated ToD effects on resting-state BOLD SD in a region of interest in the suprachiasmatic region, because this area contains the circadian pacemaker, the suprachiasmatic nucleus. The region of interest was defined using a published sphere[20] and served for a small volume correction in the BOLD SD voxel-wise group analysis. We tested for a main effect of ToD using an F-contrast with a statistical threshold of $P < 0.05$, small volume corrected.

**Analysis of visual detection task fMRI data**. The same data acquisition, pre-processing and single-subject statistical modeling parameters as for the resting-state BOLD SD analyses were applied to obtain voxel-wise BOLD SD for the whole brain, in each session (ToD) of the visual detection task blocks (see above). The first 5 volumes at the beginning of each block were removed to avoid carry-over effects from the previous block. Then, the three task blocks were concatenated and BOLD SD calculated separately for each ToD which resulted in a voxel-wise BOLD SD map for each ToD. Fixation blocks were not further analyzed, because the experimental conditions were similar to resting-state. Using SPM 12 for group statistical analyses, task-related BOLD SD in the visual cortex was probed for the ToD-dependent effects observed during resting-state. As with the resting-state data, task data on the two consecutive days were averaged voxel-wise at each ToD for each participant. We tested for BOLD SD reductions at 08:00 and 20:00 h compared to midday measurements (11:00, 14:00, and 17:00 h) using a one-sample t-test. The analysis was restricted to the visual cortex cluster that showed a main effect of ToD during resting-state and results were thus corrected within this mask at $P < 0.05$, FWE cluster-corrected (cluster-forming threshold $P < 0.001$, uncorrected). Values from significant clusters were averaged within each cluster and extracted for each ToD and participant separately using MarsBaR and used for subsequent analyses (see below). We ascertained the consistency of the effect by using the global conjunction hypothesis[53] of the contrasts (08:00 h separately against 11:00, 14:00, and 17:00 h) and (20:00 h separately against 11:00, 14:00, and 17:00 h).

**Comparing resting-state and task-related BOLD SD**. We tested for significant overall decreases in BOLD variance during visual detection compared to rest using dependent, two-tailed, paired samples $t$-tests in SPSS (significance threshold at $P < 0.05$, Bonferroni corrected). The peak-voxel BOLD SD values of significant clusters from the visual detection task were extracted for each ToD and participant using MarsBar, then averaged over all ToD and compared with the ToD-averaged resting-state BOLD SD values at the same coordinates.

**Visual detection task performance**. Omission errors (i.e., lapses) and reaction times for each participant and ToD were analyzed using SPSS Statistics. To test for ToD-dependent differences in visual detection mirroring the ToD-dependent changes of BOLD SD in sensory cortices, we performed a repeated measures ANOVA (ToD as independent variable with 6 levels, Mauchly's test of sphericity was $P \geq 0.05$, unless otherwise stated) with subsequent planned pairwise comparisons using dependent, two-tailed t-tests, testing the difference between 08:00 and 20:00 h and, midday measurements (11:00, 14:00, and 17:00 h), respectively (significance threshold set at $P < 0.05$).

**Correlation analysis of omission errors and resting-state BOLD SD**. A Pearson product-moment correlation analysis was performed to test for a linear relationship between resting-state BOLD SD in the visual cortex cluster and omission errors in visual detection at the same times. Since an ordinary correlation is not appropriate for repeated measures[54], a Pearson correlation coefficient accounting for the dependence among the repeated measures was calculated, which removes the variance between subjects[55]. Therefore, the resulting correlation coefficient represents the association between BOLD SD and omission errors over all ToD for all participants, corrected for interindividual differences. The significance threshold was set at $P < 0.05$.

**Correlation analysis of omission errors and task-related BOLD SD**. Using the same procedure as above, we also tested for a linear relationship between task-related BOLD SD in visual cortex and omission errors in visual detection at the same times of day. Since the analysis of task-related BOLD SD resulted in two significant clusters in the visual cortex, correlations were calculated for BOLD SD in the two clusters separately, significance threshold $P < 0.05$.

**Power analysis for thalamic resting-state functional connectivity**. To ascertain that the negative finding in ToD-dependent thalamocortical connectivity as an objective marker of vigilance did not result from insufficient statistical power, we performed a power analysis using a bootstrapping method on a published dataset in which vigilance changes have been investigated[18]. For this re-analysis we used exactly the same parameters that we applied in our study here and investigated whether vigilance changes (N1 sleep stage vs. wakefulness) could be detected in 14 subjects in the previous dataset. In detail, a total of 93 continuous ($\approx$3.5 min) epochs of wakefulness and 54 of N1 sleep were selected from Tagliazucchi and Laufs[18]. Two sets of 14 epochs were randomly selected (with replacement) from the wakefulness and N1 sleep sets and the voxel-wise thalamic functional connectivity (as determined by the AAL atlas[47]) was computed. We tested for significant group differences using a mass-univariate two-sample and two-tailed Student's $t$-test, thresholded at $P < 0.001$ cluster-forming threshold and $P < 0.05$, FWE corrected on the cluster level. For each iteration of the randomly selected 14 wakefulness/ N1 sleep epochs, voxels were flagged as significant if their associated $P$-value fulfilled these criteria. This was repeated for 1000 iterations, and a voxel-wise map, showing the proportion of times a voxel was deemed significant, was obtained. Since significant clusters were observed, the group size of 14 participants as used in our study can be judged sufficiently large to be sensitive to detect vigilance changes.

**Data availability**. The imaging data that support the findings of this study are available at G-NODE with identifier https://doi.org/10.12751/g-node.16591b. The code to calculate voxel-wise variables of interest in fMRI data is available upon demand.

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

## Acknowledgements

This study was supported by the August Scheidel Foundation. C.A.K. is supported by an Emmy Noether grant of the German Research Foundation (KE 1514/2-1).

## Author contributions

C.A.K. and J.H.S. conceptualized the study; L.C. and C.H. acquired the data; L.C., E.T., C.V., C.H., and C.A.K analyzed data; T.R. provided code for analysis; J.H.S. and C.A.K. provided the funding for the work; L.C., E.T., C.V., J.H.S. and C.A.K wrote the manuscript.

## Additional information

**Competing interests:** The authors declare no competing interests.

