## [Peer Review File(PDF 757 kb) · Nature Communications]

Reviewers' comments:

Reviewer #1 (Remarks to the Author):

In this manuscript, the authors describe the diurnal modulation of BOLD variance both during resting (n=14) and during a visual detection task (n=9) as well as the association between BOLD variance specifically in the visual cortex and omission errors in the visual detection task. The study protocol consist of twelve scanning session at fixed times during two subsequent days. The authors find BOLD variance in sensory cortices during resting reduced in the morning and evening compared to the afternoon. Focusing on the visual cortex only, the same diurnal pattern could be confirmed in BOLD variance during performance. As both BOLD variance in the visual cortex during resting and task performance correlated positively with omission errors, the authors conclude having identified an anticipatory mechanism which compensates for reduced signal quality at dawn and dusk.

The novelty and major strength of the paper is based on the analyses of BOLD variance (in contrast to mean BOLD signal) and the combined analyses of both resting state and task performance in the same participants at six different times of day. Identifying mechanisms underlying close-to-threshold performance is not only of great interest in a chronobiological context but also of relevance regarding processes of health, disease, and aging. The manuscript is clearly written and the methods section provides sufficient details to reproduce the work.

However, I have some concerns regarding the design of the study and the analyses of the data, which - if not addressed - weaken the conclusion of the authors considerably.

Major:

- The authors conclude an endogeneous potentially circadian regulation (p. 6, line 103-105) based on their observation that 1) masking factors can be excluded, 2) thalamo-cortical connectivity does not show a significant ToD-dependent modulation, and 3) subjective sleepiness as measured by the ESS did not significantly correlate over the six different scanning session with BOLD variance in the visual cortex.

o 1) There is a wealth of evidence that sleep and wakefulness are regulated by a complex interplay of circadian and sleep homeostatic processes, as mirrored for instance in the course of subjective sleepiness, performance, and BOLD activity over the 24-hour cycle. The study protocol used does not allow disentangling the influence of these key processes in BOLD variance or also on thalamo-cortical activity (as would be possible for instance in a forced desynchrony study, which is of course much more time-consuming and difficult to conduct). Beside sleep pressure and its interaction with circadian mechanisms, there is a range of other masking factors, which may have influenced the diurnal variation in the data, such as ambient temperature, artificial lighting, food intake, body posture, social activities, or stimulant intake (for instance caffeine, nicotine). From my perspective, it is premature to assume that an influence of both sleep pressure per se and its interaction with circadian factors on BOLD variance is unlikely. Furthermore it is not appropriate to conclude an endogenous - potentially circadian - regulation (p.5, line 94 and p.6, line 105). Similar to the section on p.9, line 180-185, about biological mechanisms underlying a circadian course

of BOLD variance, the discussion should thus include a section about pathways of the sleep-homeostatic system affecting cortical activity and potential sites of interactions with the circadian drive for wakefulness. Furthermore, as the authors assessed actigraphic data and sleep diaries, it would underline the conclusions if analyses of these data do not reveal a time-of-day dependency of any assessed masking factor (such as napping, stimulant intake etc.).

o 2) From my perspective, it is an interesting finding that there was no significant diurnal modulation in thalamo-cortical connectivity, although based on a small sample size. The result might be due to the combined influence of circadian wake-promotion and homeostatic sleep-pressure on vigilance. It is unclear to me why an absent diurnal pattern of thalamo-cortical connectivity suggests an endogenous circadian modulation of BOLD activity. The authors should specify this.

o 3) The Epworth Sleepiness Scale (ESS) is not an appropriate questionnaire to assess subjective sleepiness as a dynamic state, varying according to circadian and sleep homeostatic modulations, and thus also to diurnal variations. To assess these variations in subjective sleepiness, the Karolinska Sleepiness Scale or Stanford Sleepiness Scale would have been much more appropriate. The ESS was designed to assess sleepiness as a trait or at least relatively stable pathological state and refers to situations during the usual way of life during the last weeks. Thus it is not surprising that ESS values do not correlate with the diurnal variation of BOLD SD in the visual cortex. Or did the authors modify the ESS in order to specifically assess diurnal variations of sleepiness? If not, I recommend to either delete sections reporting correlations with ESS values or to correlate only one ESS value (or the mean) with a marker of the change in BOLD variance in the respective cortical area. In the latter case, the procedure has to be discussed in terms of an association between daytime sleepiness (understood as a relatively stable state) and diurnal modulations of BOLD variance.

- To account for habituation effects data were acquired at two subsequent days (p.2, line 45). During analyses, data were averaged per time of day, most likely because habituation effects could be excluded. It would be important to note in the methods section that data (both BOLD variance and behavioral performance) were checked for habituation effects before averaging, especially because a potential habituation effect can be assumed to be strongest in the first scanning session at the first day at 8:00, thus at a key time regarding the conclusions. The scanning sessions were not exactly timed to the times of twilight. The first scanning session took place at the very end of the morning twilight. If habituation effects cannot be excluded, effects in the morning session (at 8:00) cannot be attributed to an endogenous modulation according to twilight only.

- The authors conclude that BOLD variance in the visual cortex supports visual perception according to natural lighting conditions. Given this assumption, BOLD variance during the night is supposed to be reduced compared to daytime. However, the statistics presented do not support this conclusion as neither BOLD variance at 23:00 nor performance at this time of day differs significantly from assessments at any other time of day. How do the authors reconcile the data assessed at 23:00 within the framework of BOLD variations according to natural lighting conditions?

- I assume that diurnal variations in BOLD SD during task performance were assessed within a block design contrasting BOLD variance during rest (fixation blocks) against BOLD variance during presentation of crosshairs. Please specify the design in the methods section. Furthermore, to conclude a decrease in perception-related BOLD variance in the morning and evening, it is important to present evidence that this decrease is due to a decrease in BOLD variance during presentations of crosshairs and not to an increase of BOLD variance during fixation blocks. I furthermore suggest to include a figure (for instance, a figure 3b) depicting the time course of BOLD SD (as difference between crosshairs and fixation blocks) during task performance in the two areas of the visual cortex .

- With $n=14$ for resting state data and $n=9$ for task performance, the sample size is low. Please briefly mention the reason (i.e., sample size calculation) on which sample size was based on.

Minor:

- On page 6, line 114-120, it is misleading to use the terms "improvements" or "reductions" because "absolute" values, not difference measures, were correlated with BOLD variance. Please be more precise (for instance: "Good visual detection performance coincided with low resting-state BOLD variance).

- Please discuss briefly why omission errors were modulated by time of day (and correlated with BOLD variance) while reaction times did not follow a time-of-day dependent pattern.

- Please be more precise on p.6, line 110-112: Performance at 8:00 did significantly differ from performance at 14:00 only.

- P. 15, line 326-335: Please specify if clusters in the visual and temporal cortices were considered as one for analyses of BOLD SD values.

- P.15, line 351-354: Please specify why two analyses have been computed separately for the morning and evening. Was there a reason to differentiate between morning and evening?

Reviewer #2 (Remarks to the Author):

In the current study Cordani and colleagues investigate the much-neglected topic of circadian variations, and the work is especially strong as it investigates such (co)variations in brain function and behavior in conjunction. A group of 14 males (no females to avoid impact of menstrual cycles) was scanned six times over the course of the day (repeated on two consecutive days). Repeated-measures one-way ANOVAs showed an impact of time of day on BOLD signal variance at resting state in sensory cortices, and on omission errors in a visual detection task (9 subjects). Post-hoc t-tests showed that these neural and behavioral

measures were both reduced at dawn and dusk. Additionally, BOLD signal variance was reduced in a visual cortex region-of-interest between twilight (dawn/dusk) and other times of day (paired t-test in 9 subjects). Importantly, to show a direct relation between BOLD signal variance and behavioral performance across times of day, the authors additionally performed a correlation analysis.

The unique experimental design gives rise to a very informative study, revealing the functional importance of spontaneous brain activity. Specifically, in the understudied context of time-of-day variations, the authors interpret the observed reduction of BOLD signal variance as a mechanism to improve SNR and the detection of weak visual signals at times of twilight. Sufficient details on data acquisition and analyses have been provided to allow for reproducing the work. The novelty of the study makes it a very interesting contribution for a wide range of readership. Minor questions/suggestions follow:

- Head motion heavily contributes to BOLD signal variance. The observation that time-of-day related changes in BOLD variance are spatially restricted to sensory areas speaks against this confound carrying the effects. And I realize that 6 rigid body head motion parameters, as well as CSF/WM compartment signals capturing some of motion-related effects have been regressed out. However, I would strongly suggest that a potential contribution of head motion is explicitly tested similar to what has been done for the cardiovascular parameters. Variance and amplitudes of the motion parameters, as well number of motion outliers per scan session could be investigated.
- A brief discussion should be provided regarding sample size with respect to the fact that the visual task was available in only 9 of the subjects, a potential weakness counteracted by the multi-session aspect of the design.
- I find the investigation of direct correlations between the BOLD and behavioral measures (Fig. 2) key to the interpretations. Therefore, I suggest including a short phrase in the figure legend to clarify upfront that subject-effects were modeled appropriately. This issue of repeated measures in a correlation is explained in the methods, but hinting at it will avoid red flags as the reader goes through the manuscript core.
- Methods page 14 line 316: By averaging 'data' you are referring to the variability measures (SD, ALFF etc.) I assume?

Reviewer #3 (Remarks to the Author):

The manuscript of Kell and collaborators is original and interesting. They examined the neural bases of time of day-dependant modulation in human perception using fMRI during resting-state and close-to-threshold visual perception. They showed that resting-state signal variance dropped at times coinciding with dawn and dusk in sensory cortices only. In parallel, they found that perception-related signal variance in visual cortices decreased and

correlated negatively with detection performance.

They interpret these findings as an anticipatory mechanism that compensates for the deteriorated visual signal quality at dawn and dusk and that decreases in spontaneous neural activity improve close-to-threshold perception.

Authors could have anticipated a role for the suprachiasmatic nucleus since they acknowledge that « Animal models revealed ToD-dependent changes in genome readout, protein dynamics, and in electrophysiology that are regulated by the circadian system ». It should have been important to focus also on deeper structures and highlight the potential implication of the suprachiasmatic nucleus (Schmidt et al., Science. 2009 Apr 24;324(5926):516-9)

They excluded in their study the extreme chronotype but did not excluded other interindividual differences. Authors included young persons and frequently this population is in sleep debt. While a control with actimetry has been done for the 5 previous days before the fMRI acquisition they did not examined the impact of sleep pressure due to sleep debt. Other interindividual differences either internal (sensitivity to light through intrinsically photosensitive retinal ganglion cells, e.g. Higushi et al. PLoS One. 2013;8(3):e60310) or external (duration of smartphone exposure or other exposition to light emitting devices, e.g. Heo et al. J Psychiatr Res. 2017 Apr;87:61-70) should have been controlled.

While the authors interpret their finding as anticipatory mechanisms they suggest a causality in their observations. They highlighted correlations and did not conducted any dynamic causal modelling experiment that could model a potential directional impact between light, detection performance and signal variance in visual cortices. So far, what authors name anticipatory mechanisms and compensation mechanisms might be correlations, either cause or consequence, trigger or adaptation. A more nuanced position should be adopted.

We thank the Editor and the three Reviewers for their very constructive feedback. Please find our answers below.

Answers to the Editor

More specifically, we would like to draw your attention to Reviewer 1's concerns regarding the other factors that could have influenced the diurnal variation in the data. While we realize that it is hard to disentangle these effects, we ask that you do your best by including analyses examining interindividual differences (as requested by Reviewer 3), using dynamic causal modeling to investigate how these different variables interact, and by thoroughly discussing the results and caveats of the study.

Answer: We thank you and the reviewers for this valuable suggestion. We now included several inter-individual and within-subject factors that could have potentially influenced findings in our analyses. Besides heart rate, body temperature, breathing rate, and subjective sleepiness we added amplitude and variance of movement as well as the amount of movement outliers during scanning for each fMRI-session as possible confounders and show that the main effect of ToD is robust against their inclusion in a linear mixed model. Based on the very helpful remarks of the reviewers, we now also study whether chronotype, as a proxy for circadian phase, as well as sleep pressure and sleep debt, two homeostatic factors, and scanning day explain part of the observed ToD effect in BOLD SD. These new analyses added important new insights, because they show that the main effect of ToD remained significant when including interactions between these factors and ToD into the linear mixed model. Yet, there was also an interaction between BOLD SD with both, chronotype, and sleep pressure, as predicted by the reviewers. Because our study was not originally designed to dissociate the specific contributions of circadian and homeostatic processes, we do not over-interpret this finding and relate to previous literature.

Dynamic causal modeling requires time series for the estimation of dynamic interactions between factors. Unfortunately, such an analysis cannot be performed on single data points like the standard deviation that was the measure of interest, showing ToD effects in our study. We are confident that the statistical approach that we used here to investigate both, intra- and inter-individual factors answers correctly the questions that you and the reviewers raised.

In addition, please note that we followed all the reviewers' suggestions regarding the way we discuss our findings (see below).

Finally, I would like to point out that we are committed to publishing robust results and so take the concern regarding the sample size (raised by Reviewers 1 and 2) and ask that you provide the appropriate analyses (rather than citing previous studies) to justify this size (or add more data to ensure that the sample size is big enough to draw robust conclusions).

Answer: This is indeed an important point, particularly regarding the negative finding of ToD effects on thalamo-cortical connectivity as a proxy for vigilance. While the BOLD SD reductions at times of twilight represent strong effects (Cohen's $d > 0.8$, please see Supplementary Table 2) and were not affected by inclusion of the factor scanning day into our model that was based on data from the two scanning days, separately (see answer to Reviewer 1), we did not find any significant ToD effect on thalamo-cortical functional connectivity. To exclude that this negative finding resulted from a too small

sample size, we ran a bootstrap analysis in a published dataset (Ref. 18; Tagliazucchi and Laufs, Neuron 82, 695-708 (2014)) that included awake and dozing stages. We used exactly the same parameters that we applied in our study and investigated whether vigilance changes (N1 sleep stage vs. wakefulness) could be detected in 14 subjects in the previous dataset. In detail, a total of 93 continuous (≈ 3.5 min) independent epochs of wakefulness and 54 of N1 sleep were selected from Ref. 18. Two sets of 14 epochs were randomly selected (with replacement) from the wakefulness and N1 sleep sets and the voxel-wise thalamic functional connectivity (as determined by the AAL atlas, Ref. 47) was computed. The presence of significant differences between groups was assessed using a mass-univariate two-sample and two-tailed Student's t-test, thresholded at $p < 0.001$ cluster-forming threshold and $p < 0.05$, FWE corrected on the cluster level. For each iteration of the randomly selected 14 wakefulness/N1 sleep epochs, voxels were flagged as significant if their associated p -value fulfilled these criteria. This was repeated for 1000 iterations, and a voxel-wise map, showing the proportion of times a voxel was deemed significant, was obtained. This map is shown in the Figure below, thresholded at 80% (i.e. only those voxels that showed significant differences for more than 80% of the 1000 iterations are shown).

Indeed, the vigilance changes that could be of interest in our study and characterize the N1 sleep stage gave rise to a number of significant clusters in the expected brain regions. Please see the Figure below. We feel that this demonstrates that our sample is large enough to interpret our negative finding.

Figure legend: Clusters showing significant, vigilance-related differences in thalamic functional connectivity (wakefulness vs. N1 sleep, $n=14$ randomly selected with replacement over 1000 iterations, $p < 0.001$ cluster-forming threshold, and $p < 0.05$ FWE correction on the cluster level) for at least 80% of the iterations in the dataset of Ref. 18; Tagliazucchi and Laufs, Neuron 82, 695-708 (2014).

Therefore, we have changed the manuscript accordingly:

“Because the main effect of ToD in BOLD SD remained significant when adjusting for subjective sleepiness, it is likely that the ToD-dependent modulation is not explained by vigilance states. We additionally investigated whether an objective marker of EEG-documented vigilance changes^{18, 19} showed comparable ToD-effects. We tested the main effect of ToD in resting-state thalamo-cortical connectivity using correlation maps of thalamic BOLD time series with all other brain voxels. This parameter was not affected by ToD (repeated measures ANOVA, cluster size $k = 3$, $P > 0.999$, family-wise error corrected, FWE) suggesting that both subjective and objective measures of vigilance states do not explain the diurnal modulation in resting-state BOLD SD in the sensory cortices. To ascertain that this negative finding did not result from insufficient statistical power, we performed a bootstrap analysis using a published dataset in which vigilance changes have been investigated¹⁸, testing whether a group size of 14, as used in the present study, is sufficient to detect vigilance changes. Indeed, vigilance-related changes in thalamo-cortical connectivity can reliably ($1 - \beta = 0.8$) be detected in sample sizes as the one used in this study.”

In the methods section we added:

“Power analysis for thalamic resting-state functional connectivity

*To ascertain that the negative finding in ToD-dependent thalamocortical connectivity as an objective marker of vigilance did not result from insufficient statistical power, we performed a power analysis using a bootstrapping method on a published dataset in which vigilance changes have been investigated¹⁸. For this re-analysis we used exactly the same parameters that we applied in our study here and investigated whether vigilance changes (N1 sleep stage vs. wakefulness) could be detected in 14 subjects in the previous dataset. In detail, a total of 93 continuous (≈ 3.5 min) epochs of wakefulness and 54 of N1 sleep were selected from Tagliazucchi and Laufs¹⁸. Two sets of 14 epochs were randomly selected (with replacement) from the wakefulness and N1 sleep sets and the voxel-wise thalamic functional connectivity (as determined by the AAL atlas⁴⁷) were computed. We tested for significant group differences using a mass-univariate two-sample and two-tailed Student’s *t*-test, thresholded at $P < 0.001$ cluster-forming threshold and $P < 0.05$, FWE corrected on the cluster level. For each iteration of the randomly selected 14 wakefulness/N1 sleep epochs, voxels were flagged as significant if their associated *P*-value fulfilled these criteria. This was repeated for 1000 iterations, and a voxel-wise map, showing the proportion of times a voxel was deemed significant, was obtained. Since significant clusters were observed, the group size of 14 participants as used in our study can be judged sufficiently large to be sensitive to detect vigilance changes.”*

Answers to Reviewer #1

We appreciate very much that Reviewer #1 states that our presented data bear 'novelty' and have 'major strength'. He/She rates identified mechanisms as 'of great interest in a chronobiological context but also of relevance regarding processes of health, disease, and aging' and that 'the manuscript is clearly written and the methods section provides sufficient details to reproduce the work'.

Major:

- The authors conclude an endogenous potentially circadian regulation (p. 6, line 103-105) based on their observation that 1) masking factors can be excluded, 2) thalamo-cortical connectivity does not show a significant ToD-dependent modulation, and 3) subjective sleepiness as measured by the ESS did not significantly correlate over the six different scanning session with BOLD variance in the visual cortex.

o 1) There is a wealth of evidence that sleep and wakefulness are regulated by a complex interplay of circadian and sleep homeostatic processes, as mirrored for instance in the course of subjective sleepiness, performance, and BOLD activity over the 24-hour cycle. The study protocol used does not allow disentangling the influence of these key processes in BOLD variance or also on thalamo-cortical activity (as would be possible for instance in a forced desynchrony study, which is of course much more time-consuming and difficult to conduct).

Answer: We did not intend to convey the message that sleep homeostatic processes do not play a role in diurnal modulation of brain activity. With our analyses of thalamo-cortical connectivity and the subjective sleepiness reports we studied whether vigilance states - as a result from interactions between circadian and homeostatic factors - follow the diurnal modulation that BOLD SD revealed. Because no such relationship was found (this negative finding is now backed up by the aforementioned power calculations) we are confident that vigilance state alone does not explain our results. This does not exclude effects of homeostatic processes per se. Based on the reviewer's suggestion, we now explicitly investigate whether the main effect of ToD in BOLD SD remained significant when including chronotype, sleep pressure, and sleep debt (see also answer to Reviewer 3) as factors and their interaction with ToD in our new linear mixed model (that also accounted for the confounding factors, see also next point). While the main effect of ToD remained highly significant, the interactions revealed that the modulation of BOLD SD could indeed at least in part be explained by chronotype, a variable that is related with the circadian system, and sleep pressure, a homeostatic parameter. We therefore have now changed the manuscript accordingly:

"Because the main effect of ToD in BOLD SD remained significant when adjusting for subjective sleepiness, it is likely that the ToD-dependent modulation is not explained by vigilance states. The lack of an association with vigilance state rather suggests an endogenous modulation of resting state cortical activity by circadian and/or homeostatic factors. ... Indeed, both, chronotype, a proxy for circadian phase, and sleep pressure, a homeostatic factor, explained part of the ToD modulation in BOLD variance. The linear mixed model on resting-state BOLD SD in the visual cortex showed a significant interaction between ToD and chronotype ($F(5, 34.603) = 2.738, P = 0.037$) and between ToD and sleep pressure ($F(5, 32.21) = 5.128, P = 0.001$). There was no significant interaction between ToD and sleep debt ($F(5, 38.554) = 1.922, P = 0.113$).

While our study was not originally designed to dissociate the specific contribution of circadian and homeostatic factors in the ToD-dependent modulation of sensory cortex activity, a forced desynchrony study in a larger sample documented interactions between circadian and homeostatic influences and BOLD activity during a visual psychomotor vigilance task¹⁰. In this publication, an interaction between homeostatic and circadian factors was particularly observed in posterior, mostly occipital cortices, while the circadian modulation seemed less important in the frontal cortices. This observation suggests that the circadian system modulates perceptual processes in the sensory cortices and could potentially be the primary mediator of the ToD effects observed here.”

Beside sleep pressure and its interaction with circadian mechanisms, there is a range of other masking factors, which may have influenced the diurnal variation in the data, such as ambient temperature, artificial lighting, food intake, body posture, social activities, or stimulant intake (for instance caffeine, nicotine). From my perspective, it is premature to assume that an influence of both sleep pressure per se and its interaction with circadian factors on BOLD variance is unlikely. Furthermore it is not appropriate to conclude an endogenous - potentially circadian - regulation (p.5, line 94 and p.6, line 105). Similar to the section on p.9, line 180-185, about biological mechanisms underlying a circadian course of BOLD variance, the discussion should thus include a section about pathways of the sleep-homeostatic system affecting cortical activity and potential sites of interactions with the circadian drive for wakefulness. Furthermore, as the authors assessed actigraphic data and sleep diaries, it would underline the conclusions if analyses of these data do not reveal a time-of-day dependency of any assessed masking factor (such as napping, stimulant intake etc.).

Answer: We now added the requested information by estimating a linear mixed model that included possible masking factors. Importantly, the main effect of ToD in BOLD SD was robust even when including these potential confounders and interactions with chronotype and homeostatic factors into the linear mixed model. As predicted by the reviewer, chronotype as well as sleep pressure explains part of the ToD-dependent modulation in BOLD SD:

We now write in our manuscript:

“A range of masking factors could have potentially influenced our observation. Yet, the ToD modulation of BOLD SD in the visual cortex remained significant after accounting for potential confounders, including heart rate, breathing rate, body temperature, subjective sleepiness, and amplitude and variance of head motion parameters, and the number of outliers in head motion parameters in a linear mixed model (main effect of ToD: $F(5, 34.029) = 4.177, P = 0.005$). This model also included chronotype, sleep pressure, sleep debt, and scanning days to investigate effects of interindividual differences (see below) and habituation on resting-state BOLD SD. The repetitive measurements on two subsequent scanning days did not introduce habituation effects, because ToD remained significant when accounting for scanning days in the model and there was no significant interaction between ToD and scanning day ($F(5, 41.829) = 0.871, P = 0.509$).”

and

“Participants did not nap or consume stimulants during scanning days.”

We also added in the methods section:

“Multi-variable adjustment of linear mixed models

Resting-state activity can be affected by a range of potential confounders. To test whether the observed ToD effects in BOLD SD were driven by such effects, we calculated a linear mixed model with resting-state BOLD SD in the visual cortex as dependent variable. This model contained separate data points for the two scanning days and included ToD and scanning day as repeated measures fixed effects (autoregressive covariance structure AR(1)), as well as heart rate, breathing rate, body temperature, subjective sleepiness, amplitude and variance of head motion parameters, and the number of outliers in head motion parameters (defined as values above the mean $\pm 2,5$ standard deviations), chronotype, sleep pressure, and sleep debt as fixed effects. Subjects were included as a random effects variable. We used the restricted maximum likelihood estimation method and tested whether the main effect of ToD in BOLD SD in the visual cortex remained significant ($P < 0.05$), when including all these potential confounding factors. To study whether chronotype, sleep pressure, and sleep debt as well as scanning day explained part of the ToD modulation in BOLD variance, we additionally included multiplicative interaction terms in the multi-variable adjusted, linear mixed model. Interactions were considered significant at $P < 0.05$. A significant interaction between ToD and scanning day could potentially reveal habituation effects due to repetitive measurements on two subsequent scanning days. Yet, the interaction was not significant ($F(5, 41.829) = 0.871, P = 0.509$).”

Room temperature in the scanner was kept constant but outside temperature was not measured. However, body temperature did not affect the main effect of ToD in BOLD SD in the linear mixed models. Light exposure and calorie intake was not registered, which is a limitation of the study, which is now mentioned more clearly:

“In addition, we did not measure light exposure prior to scanning and calorie intake, so that we could not examine the relationship between these factors and BOLD SD. Note that previous light exposure and calorie intake must have been very different at 08:00 h and 20:00 h, time points at which BOLD SD in sensory cortices was equally reduced.”

However, the diaries did not reveal any consistent relationship between time of day and body posture or social interactions but, participants were not explicitly asked to report these factors.

o 2) From my perspective, it is an interesting finding that there was no significant diurnal modulation in thalamo-cortical connectivity, although based on a small sample size. The result might be due to the combined influence of circadian wake-promotion and homeostatic sleep-pressure on vigilance. It is unclear to me why an absent diurnal pattern of thalamo-cortical connectivity suggests an endogenous circadian modulation of BOLD activity. The authors should specify this.

Answer: The claim that the modulation in BOLD SD occurs endogenously rests upon the observation of such changes during resting state in addition to a similar task-related modulation. “Endogenous” in that sense implies that measured changes do not result from stimulus features or ambient lighting conditions. We thus specified: *“Because the ToD-dependent modulation of BOLD SD represents changes in the spontaneous neural activity during resting-state, independent of sensory stimulation and task demands, our results suggest a ToD-dependent endogenous regulation of cortical activity.”*

We only suggest that the circadian system is an important source of time information for such endogenous anticipation and now provide some empirical results supporting this

conjecture (please see our answers above). We apologize if our previous version of the manuscript implied that homeostatic factors do not contribute to ToD dependent modulation of brain activity. We now report an interaction between circadian and homeostatic factors that modulates resting state BOLD SD and discuss more cautiously. See, e.g.: *“Animal models revealed ToD-dependent changes in genome readout, protein dynamics, and in electrophysiology that are regulated by the circadian system^{3, 4, 5} and homeostatic factors⁶.”*

and

“The lack of an association with vigilance state rather suggests an endogenous modulation of resting state cortical activity by circadian and/or homeostatic factors.”

and

“While our study was not originally designed to dissociate the specific contribution of circadian and homeostatic factors in the ToD-dependent modulation of sensory cortex activity, a forced desynchrony study in a larger sample documented interactions between circadian and homeostatic influences and BOLD activity during a visual psychomotor vigilance task¹⁰.”

A potential change in thalamo-cortical connectivity could have been expected at 23:00 when participants reported increased sleepiness. The bootstrap analysis demonstrates that our sample size was sufficiently large to possibly detect vigilance changes. Yet, there was no significant ToD effect on thalamo-cortical connectivity in our data detectable.

o 3) The Epworth Sleepiness Scale (ESS) is not an appropriate questionnaire to assess subjective sleepiness as a dynamic state, varying according to circadian and sleep homeostatic modulations, and thus also to diurnal variations. To assess these variations in subjective sleepiness, the Karolinska Sleepiness Scale or Stanford Sleepiness Scale would have been much more appropriate. The ESS was designed to assess sleepiness as a trait or at least relatively stable pathological state and refers to situations during the usual way of life during the last weeks. Thus it is not surprising that ESS values do not correlate with the diurnal variation of BOLD SD in the visual cortex. Or did the authors modify the ESS in order to specifically assess diurnal variations of sleepiness? If not, I recommend to either delete sections reporting correlations with ESS values or to correlate only one ESS value (or the mean) with a marker of the change in BOLD variance in the respective cortical area.

In the latter case, the procedure has to be discussed in terms of an association between daytime sleepiness (understood as a relatively stable state) and diurnal modulations of BOLD variance.

Answer: We apologize for not mentioning that we asked participants to rate their current vigilance state in the ESS by asking them to project themselves at each measurement in the questionnaire situations and rate their actual sleepiness.

We now write in the method section of our manuscript:

“Before each fMRI session, body temperature was measured with an ear thermometer and the sleep propensity of the participants was assessed with a self-report questionnaire based on a modified version of the Epworth Sleepiness Scale (ESS)

questionnaire⁴⁴, in which participants were asked to rate their sleep propensity by projecting their actual vigilance state in the situations provided by the questionnaire to quantify diurnal sleepiness.”

- To account for habituation effects data were acquired at two subsequent days (p.2, line 45). During analyses, data were averaged per time of day, most likely because habituation effects could be excluded. It would be important to note in the methods section that data (both BOLD variance and behavioral performance) were checked for habituation effects before averaging, especially because a potential habituation effect can be assumed to be strongest in the first scanning session at the first day at 8:00, thus at a key time regarding the conclusions. The scanning sessions were not exactly timed to the times of twilight. The first scanning session took place at the very end of the morning twilight. If habituation effects cannot be excluded, effects in the morning session (at 8:00) cannot be attributed to an endogenous modulation according to twilight only.

Answer: This is an important point. We now added scanning day as a repeated measure fixed effect in the linear mixed model in which data points from the two scanning days were entered separately and studied whether inclusion of this factor as well as its interaction with ToD rendered the main effect of ToD non-significant. Clearly, averaging time points did not alter the modulation by ToD.

We now write in our manuscript:

“The repetitive measurements on two subsequent scanning days did not introduce habituation effects, because ToD remained significant when accounting for scanning days in the model and there was no significant interaction between ToD and scanning day ($F(5, 41.829) = 0.871, P = 0.509$).”

And in the methods section, we now write: *“To study whether chronotype, sleep pressure, and sleep debt as well as scanning day explained part of the ToD modulation in BOLD variance, we additionally included multiplicative interaction terms in the multi-variable adjusted, linear mixed model. Interactions were considered significant at $P < 0.05$. A significant interaction between ToD and scanning day could potentially reveal habituation effects due to repetitive measurements on two subsequent scanning days. Yet, the interaction was not significant ($F(5, 41.829) = 0.871, P = 0.509$).”*

- The authors conclude that BOLD variance in the visual cortex supports visual perception according to natural lighting conditions. Given this assumption, BOLD variance during the night is supposed to be reduced compared to daytime. However, the statistics presented do not support this conclusion as neither BOLD variance at 23:00 nor performance at this time of day differs significantly from assessments at any other time of day. How do the authors reconcile the data assessed at 23:00 within the framework of BOLD variations according to natural lighting conditions?

Answer: As we did not investigate time points during nighttime, we cannot make any claim regarding the modulation of BOLD SD during nighttime. One could speculate that variance is only suppressed during activity phases in twilight or darkness, because BOLD SD actually increases during sleep compared to wakefulness in sensory cortices (see Ref. 18; Tagliazucchi and Laufs, *Neuron* 82, 695-708 (2014)).

We now added this sentence to the discussion:

“Because we did not investigate time points during nighttime, we cannot clarify whether the observed BOLD variance reductions occur isolated at times of twilight or whether they also occur during later phases of the night.”

- I assume that diurnal variations in BOLD SD during task performance were assessed within a block design contrasting BOLD variance during rest (fixation blocks) against BOLD variance during presentation of crosshairs. Please specify the design in the methods section. Furthermore, to conclude a decrease in perception-related BOLD variance in the morning and evening, it is important to present evidence that this decrease is due to a decrease in BOLD variance during presentations of crosshairs and not to an increase of BOLD variance during fixation blocks. I furthermore suggest to include a figure (for instance, a figure 3b) depicting the time course of BOLD SD (as difference between crosshairs and fixation blocks) during task performance in the two areas of the visual cortex.

Answer: The analysis of BOLD SD in the visual detection task was performed on the task blocks only, using the same preprocessing and statistical modeling parameters as for the resting-state analysis. BOLD SD during task blocks was therefore not contrasted against fixation blocks. We apologize for the misunderstanding. Since we excluded the first 5 volumes at the beginning of each block to avoid contamination of BOLD activity from the previous block when calculating BOLD SD, only a total of 30 fixation block-volumes remained, which is a too small sample to calculate variance from. However, we show that there is an additional decrease of BOLD SD during fixation blocks compared to resting state using T-Tests. We clarified this in the results section:

“Task-related BOLD SD in two clusters in the primary and secondary visual cortex was reduced at 8:00 h and 20:00 h, as compared to midday measurements (Fig. 3a and b). In comparison to resting state, BOLD SD during the visual detection task was overall decreased in both the primary and secondary visual cortex (Rest-task: $t(8) = 3.82$, $P = 0.010$, Bonferroni corrected, Cohen’s $d = 0.99$, mean difference = 1.26) and the secondary visual cortex (Rest-task: $t(8) = 4.75$, $P = 0.002$, Bonferroni corrected, Cohen’s $d = 1.30$, mean difference = 1.63).”

For clarification we now have changed the method section and write:

“The same data acquisition, preprocessing and single-subject statistical modeling parameters as for the resting-state BOLD SD analyses were applied to obtain voxel-wise BOLD SD for the whole brain, in each session (ToD) of the visual detection task blocks (see above). The first 5 volumes at the beginning of each block were removed to avoid carry-over effects from the previous block. Then, the three task blocks were concatenated and BOLD SD calculated separately for each ToD which resulted in a voxel-wise BOLD SD map for each ToD. Fixation blocks were not further analyzed, because the experimental conditions were similar to resting-state. Using SPM 12 for group statistical analyses, task-related BOLD SD in the visual cortex was probed for the ToD-dependent effects observed during resting-state. As with the resting-state data, task data on the two consecutive days were averaged voxel-wise at each ToD for each participant. We tested for BOLD SD reductions at 08:00 h and 20:00 h compared to midday measurements (11:00 h, 14:00 h and 17:00 h) using a one-sample t-test. The analysis was restricted to the visual cortex cluster that showed a main effect of ToD during resting-state and results were thus corrected within this mask at $P < 0.05$, FWE cluster-corrected (cluster-forming threshold $P < 0.001$, uncorrected). Values from significant clusters were averaged within each cluster and extracted for each ToD and participant separately using MarsBaR and used for subsequent analyses (see below). We ascertained the consistency of the effect by using the global conjunction hypothesis⁵³ of the contrasts {08:00 h separately against 11:00 h, 14:00 h and 17:00 h} and {20:00 h separately against 11:00 h, 14:00 h and 17:00 h}.”

To illustrate the ToD-dependent modulation in task-related activity in the visual cortex, we now added another panel to Figure 3 that depicts the tested effect.

- With $n=14$ for resting state data and $n=9$ for task performance, the sample size is low. Please briefly mention the reason (i.e., sample size calculation) on which sample size was based on.

Answer: BOLD SD is a relatively new measure in the imaging field and unfortunately, we have not found a single study using this measure that reported effect sizes. Consequently, we could not base any power calculations on previously published data. We report the observed effect sizes (see Supplementary Table 1) that are actually large (Cohen's $d > 0.8$). This suggests a sufficiently large sample size. For the negative finding of thalamo-cortical connectivity, we now provide the details on the power calculation (see answer to the Editor). We added a sentence to the methods section:
"To enable a quantification of potential habituation effects, participants came in on two days, and were assessed multiple times throughout the day. Because of this resource- and cost-intense study design, sample size was limited."

Minor:

- On page 6, line 114-120, it is misleading to use the terms "improvements" or "reductions" because "absolute" values, not difference measures, were correlated with BOLD variance. Please be more precise (for instance: "Good visual detection performance coincided with low resting-state BOLD variance").

Answer: We have addressed the valid point made by the reviewer.

- Please discuss briefly why omission errors were modulated by time of day (and correlated with BOLD variance) while reaction times did not follow a time-of-day dependent pattern.

Answer: We now write in the manuscript:
"Reaction times were not significantly different throughout the day ($F(5, 40) = 2.01, P = 0.097$; partial eta-squared = 0.201), indicating that omission errors are a better measure for detection of brief stimuli than reaction times."

- Please be more precise on p.6, line 110-112: Performance at 8:00 did significantly differ from performance at 14:00 only.

Answer: We now reformulated the phrase and write *"Participants performed best at 08:00 h (mean \pm SD omission errors = 1.17 ± 1.17) and 20:00 h (mean \pm SD omission errors = 0.56 ± 0.88) compared to midday (omission errors all ≥ 2.00 , Supplementary Table 3)."* We now believe that the sentence does not lead anymore to misunderstandings, because 14:00 represents midday and all the statistical details are reported in the Supplementary Table 3.

- P. 15, line 326-335: Please specify if clusters in the visual and temporal cortices were considered as one for analyses of BOLD SD values.

Answer: We now write “Clusters in left and right visual cortex were also considered as one. The three clusters in the left temporal cortex were also considered as one.”

- P.15, line 351-354: Please specify why two analyses have been computed separately for the morning and evening. Was there a reason to differentiate between morning and evening?

Answer: We apologize for the misleading expression. We now write more clearly: “We tested for BOLD SD reductions at 08:00 h and 20:00 h compared to midday measurements (11:00 h, 14:00 h and 17:00 h) using a one-sample t-test. The analysis was restricted to the visual cortex cluster that showed a main effect of ToD during resting-state and results were thus corrected within this mask at $P < 0.05$, FWE cluster-corrected (cluster-forming threshold $P < 0.001$, uncorrected). Values from significant clusters were averaged within each cluster and extracted for each ToD and participant separately using MarsBaR and used for subsequent analyses (see below). We ascertained the consistency of the effect by using the global conjunction hypothesis⁵³ of the contrasts {08:00 h separately against 11:00 h, 14:00 h and 17:00 h} and {20:00 h separately against 11:00 h, 14:00 h and 17:00 h}.”

Answers to Reviewer #2:

We appreciate that reviewer #2 finds that we ‘investigate the much-neglected topic of circadian variations’ and that the work is especially strong as it investigates such (co)variations in brain function and behavior in conjunction.’ The reviewer states further that ‘The unique experimental design gives rise to a very informative study’ and that we provide ‘Sufficient details on data acquisition and analyses to allow for reproducing the work.’, and that ‘The novelty of the study makes it a very interesting contribution for a wide range of readership.’

- Head motion heavily contributes to BOLD signal variance. The observation that time-of-day related changes in BOLD variance are spatially restricted to sensory areas speaks against this confound carrying the effects. And I realize that 6 rigid body head motion parameters, as well as CSF/WM compartment signals capturing some of motion-related effects have been regressed out. However, I would strongly suggest that a potential contribution of head motion is explicitly tested similar to what has been done for the cardiovascular parameters. Variance and amplitudes of the motion parameters, as well number of motion outliers per scan session could be investigated.

Answer: We appreciate the given suggestion very much indeed. Because of the suggestions made by the other Reviewers and the Editor to include many more variables, we now refrained from testing each single parameter for a main effect of ToD and in a secondary step for correlation with BOLD SD. Instead, we now included the proposed confounders in a linear mixed model to statistically test whether the observed main effect of ToD in BOLD SD was robust. We now show that amplitude and variance of movement as well as number of outliers do not affect the main effect of ToD in BOLD SD. Please see answer to Reviewer 1.

- A brief discussion should be provided regarding sample size with respect to the fact that the visual task was available in only 9 of the subjects, a potential weakness counteracted by the multi-session aspect of the design.

Answer: We now added a sentence to the methods section:

"To enable a quantification of potential habituation effects, participants came in on two days, and were assessed multiple times throughout the day. Because of this resource- and cost-intensive study design, sample size was limited."

- I find the investigation of direct correlations between the BOLD and behavioral measures (Fig. 2) key to the interpretations. Therefore, I suggest including a short phrase in the figure legend to clarify upfront that subject-effects were modeled appropriately. This issue of repeated measures in a correlation is explained in the methods, but hinting at it will avoid red flags as the reader goes through the manuscript core.

Answer: We followed the reviewer's suggestion and write now:

"Pearson correlations, that accounted for the dependence among the repeated measures, were calculated to study the relationship between behavior and BOLD SD during rest (A) and task (B)."

- Methods page 14 line 316: By averaging 'data' you are referring to the variability measures (SD, ALFF etc.) I assume?

Answer: We now write:

"First, for each participant, the BOLD SD, ALFF, and thalamic connectivity values acquired on the two consecutive days were averaged at each ToD separately, voxel-wise using the ImCalc tool in SPM, yielding BOLD SD, ALFF and thalamic connectivity whole brain maps for each of the six time points of interest."

Answers to Reviewer #3

We are very happy that also reviewer #3 finds our manuscript 'original and interesting.'

Authors could have anticipated a role for the suprachiasmatic nucleus since they acknowledge that « Animal models revealed ToD-dependent changes in genome readout, protein dynamics, and in electrophysiology that are regulated by the circadian system ». It should have been important to focus also on deeper structures and highlight the potential implication of the suprachiasmatic nucleus (Schmidt et al., Science. 2009 Apr 24;324(5926):516-9)

Answer: We focused on the ToD modulation of cortical activity because of the known relationship between endogenous cortical activity and perception (e.g., Hesselmann et al., PNAS 105, 10984-10989 (2008)). However, the observed cortical effects are likely under subcortical, and potentially circadian, control. Based on your suggestion, we studied a ROI in the suprachiasmatic region defined by the ROIs in our Ref. 20 (Schmidt et al., Science 324, 516-519 [2009]) to search for similar BOLD SD effects. We were not

surprised that we did not find such an effect, because the physiologically relevant SCN output that codes circadian time at each ToD is mean firing rate (Ref. 21; Shibata et al., Brain Res. 247, 154-158 [1982] and Ref. 22; Meijer et al., Brain Res. 753, 322-327 [1997]), a very different measure than the variance in cortical activity that has already previously been linked with perception.

We now report in our manuscript:

“The lack of an association with vigilance state rather suggests an endogenous modulation of resting state cortical activity by circadian and/or homeostatic factors. When we examined the region of the suprachiasmatic nucleus in the hypothalamus²⁰, which is the central pacemaker of the circadian system in the brain, we did not observe a modulation of resting-state BOLD variance as a function of ToD (all voxels $P > 0.05$, small volume corrected). Still, it remains possible that the ToD effects in BOLD SD in sensory cortices are regulated by the circadian system, because the suprachiasmatic nucleus codes time at each ToD in the magnitude of firing rate rather than in firing rate variance^{21,22}. Indeed, both, chronotype, a proxy for circadian phase, and sleep pressure, a homeostatic factor, explained part of the ToD modulation in BOLD variance. The linear mixed model on resting-state BOLD SD in the visual cortex showed a significant interaction between ToD and chronotype ($F(5, 34.603) = 2.738, P = 0.037$) and between ToD and sleep pressure ($F(5, 32.21) = 5.128, P = 0.001$). There was no significant interaction between ToD and sleep debt ($F(5, 38.554) = 1.922, P = 0.113$).”

and in the methods section:

“Region of interest analysis in the suprachiasmatic region

We additionally investigated ToD effects on resting-state BOLD SD in a region of interest in the suprachiasmatic region, because this area contains the circadian pacemaker, the suprachiasmatic nucleus. The region of interest was defined using a published sphere²⁰ and served for a small volume correction in the BOLD SD voxel-wise group analysis. We tested for a main effect of ToD using an F-contrast with a statistical threshold of $p < 0.05$, small volume corrected.”

We now additionally study the interaction between ToD and the circadian and homeostatic factors using mixed linear models, please see our answers to the Editor, Reviewer 1 and below.

They excluded in their study the extreme chronotype but did not excluded other interindividual differences. Authors included young persons and frequently this population is in sleep debt. While a control with actimetry has been done for the 5 previous days before the fMRI acquisition they did not examined the impact of sleep pressure due to sleep debt. Other interindividual differences either internal (sensitivity to light through intrinsically photosensitive retinal ganglion cells, e.g. Higushi et al. PLoS One. 2013;8(3):e60310) or external (duration of smartphone exposure or other exposition to light emitting devices, e.g. Heo et al. J Psychiatr Res. 2017 Apr;87:61-70) should have been controlled.

Answer: Based on the reviewer’s suggestion, we now added sleep debt together with sleep pressure as homeostatic factors and chronotype as a circadian factor to our analyses. Indeed, these factors explained part of the resting state BOLD SD. Please see our answer to Reviewer 1.

Unfortunately, our sample size is too small to study genetic polymorphisms. Light exposure was not registered, which is a limitation of the study, which is now mentioned more clearly:

"In addition, we did not measure light exposure prior to scanning and calorie intake, so that we could not examine the relationship between these factors and BOLD SD. Note that previous light exposure and calorie intake must have been very different at 08:00 h and 20:00 h, time points at which BOLD SD in sensory cortices was equally reduced."

While the authors interpret their finding as anticipatory mechanisms they suggest a causality in their observations. They highlighted correlations and did not conducted any dynamic causal modelling experiment that could model a potential directional impact between light, detection performance and signal variance in visual cortices. So far, what authors name anticipatory mechanisms and compensation mechanisms might be correlations, either cause or consequence, trigger or adaptation. A more nuanced position should be adopted.

Answer: While we unfortunately cannot perform dynamic causal modeling on our data (single data points of standard deviation and, consequently, no time series data; no strong sensory input to model), we now discuss more clearly, what we mean by anticipatory. Because we observe a strong modulation of BOLD SD in sensory cortices during resting state, we can assume that these effects are not state-dependent in the sense that they are neither induced by task demands nor by sensory stimulation. Improvement in behavior during a later task must therefore represent a consequence and not a cause of variance suppression. Because lighting conditions were always identical during scanning sessions and because we controlled for a broad range of potential masking factors, we are confident to have revealed an endogenous, anticipatory mechanism. Please see also our previous answers. Nevertheless, we now discuss this in more detail:

"Because the ToD-dependent modulation of BOLD SD represents changes in the spontaneous neural activity during resting-state, independent of sensory stimulation and task demands, our results suggest a ToD-dependent endogenous regulation of cortical activity. ...

This relationship between behavior during the visual detection task and prior task-independent resting-state activity points towards an anticipatory mechanism that facilitates visual perception. ...

Neuromodulatory systems such as the acetylcholine-basal forebrain system or the norepinephrine-locus coeruleus system are known to be under circadian influence^{23, 24} and to modulate homeostatic processes⁶. Both noradrenergic and cholinergic systems impact neural processing in sensory cortices by increasing the signal to noise ratio (SNR) and consequently facilitate processing of weak incoming signals^{25, 26}. Neuromodulators could thus potentially mediate both, the SNR enhancements and ToD-dependency of endogenous resting-state and perception-related BOLD activity. ...

In addition, we did not measure light exposure prior to scanning and calorie intake, so that we could not examine the relationship between these factors and BOLD SD. Note that previous light exposure and calorie intake must have been very different at 08:00 h and 20:00 h, time points at which BOLD SD in sensory cortices was equally reduced."

REVIEWERS' COMMENTS:

Reviewer #1 (Remarks to the Author):

I thank the authors for carefully addressing my scientific concerns and correcting my interpretations. Only two points remain.

The authors incorrectly describe the paper of Muto et al. (2016) as a forced desynchrony study, which however was a classical sleep deprivation protocol. Please correct this.

The authors state that they did not investigate BOLD SD during nighttime. Thus I assume that they do not consider the scan at 23:00 to have taken place at night. However, participants were instructed to follow regular bedtimes between 23:00 and 00:00 during the week before scanning so that they were most likely scanned after passing the dim-light melatonin onset, usually considered as start of the biological night. I thus suggest changing the statement that there are no data acquired during the biological night.

Reviewer #2 (Remarks to the Author):

All my questions have been answered, and the suggestion of explicitly accounting for head motion confounds has been implemented appropriately. I have no further concerns or suggestions.

Reviewer #3 (Remarks to the Author):

Authors satisfactorily answered to my queries.

Response to Reviewers

Reviewer #1:

I thank the authors for carefully addressing my scientific concerns and correcting my interpretations. Only two points remain.

The authors incorrectly describe the paper of Muto et al. (2016) as a forced desynchrony study, which however was a classical sleep deprivation protocol. Please correct this.

Answer: We apologize for this mistake. We now write: *“While our study was not originally designed to dissociate the specific contribution of circadian and homeostatic factors in the ToD-dependent modulation of sensory cortex activity, a sleep deprivation study in a larger sample documented interactions between circadian and homeostatic influences and BOLD activity during a visual psychomotor vigilance task.”*

The authors state that they did not investigate BOLD SD during nighttime. Thus I assume that they do not consider the scan at 23:00 to have taken place at night. However, participants were instructed to follow regular bedtimes between 23:00 and 00:00 during the week before scanning so that they were most likely scanned after passing the dim-light melatonin onset, usually considered as start of the biological night. I thus suggest changing the statement that there are no data acquired during the biological night.

Answer: We now write: *“Because we did not investigate time points after 23:00 h, we cannot determine whether the observed BOLD variance reductions occur isolated at times of twilight or whether they also occur during later phases of the night.”*

Reviewer #2 (Remarks to the Author):

All my questions have been answered, and the suggestion of explicitly accounting for head motion confounds has been implemented appropriately. I have no further concerns or suggestions.

Reviewer #3 (Remarks to the Author):

Authors satisfactorily answered to my queries.

Answer: We thank all reviewers for their helpful contributions.